# WATT: Weight Average Test-Time Adaptation of CLIP

**David Osowiechi*** **Mehrdad Noori*** **Gustavo A. Vargas Hakim**

**Moslem Yazdanpanah** **Ali Bahri** **Milad Cheraghalikhani** **Sahar Dastani**

**Farzad Beizaee** **Ismail Ben Ayed** **Christian Desrosiers**

LIVIA, ÉTS Montréal, Canada
International Laboratory on Learning Systems (ILLS)

## Abstract

Vision-Language Models (VLMs) such as CLIP have yielded unprecedented performances for zero-shot image classification, yet their generalization capability may still be seriously challenged when confronted to domain shifts. In response, we present Weight Average Test-Time Adaptation (WATT) of CLIP, a new approach facilitating full test-time adaptation (TTA) of this VLM. Our method employs a diverse set of templates for text prompts, augmenting the existing framework of CLIP. Predictions are utilized as pseudo labels for model updates, followed by weight averaging to consolidate the learned information globally. Furthermore, we introduce a text ensemble strategy, enhancing the overall test performance by aggregating diverse textual cues. Our findings underscore the effectiveness of WATT across diverse datasets, including CIFAR-10-C, CIFAR-10.1, CIFAR-100-C, VisDA-C, and several other challenging datasets, effectively covering a wide range of domain shifts. Notably, these enhancements are achieved without the need for additional model transformations or trainable modules. Moreover, compared to other TTA methods, our approach can operate effectively with just a single image. The code is available at: `https://github.com/Mehrdad-Noori/WATT`.

## 1 Introduction

The integration of vision and language modalities into a unified learning framework, known as Vision Language models (VLM), has shown remarkable effectiveness in a broad range of vision-related tasks [1, 2, 3]. Notably, these models excel in zero-shot generalization scenarios, where they demonstrate proficiency in tasks beyond their original training scope, without requiring additional fine-tuning supervision. Applications of models like CLIP [1] extend across diverse domains including video recognition [4], audio processing [5], and medical imaging [6]. These advancements underscore the pivotal role of such methods in shaping the trajectory of future research and applications in machine learning.

Despite its powerful capabilities, CLIP, like other traditional deep architectures such as Convolutional Neural Networks (CNNs), experiences performance degradation when confronted with domains it has not been trained on. Current research trends emphasize the importance of domain adaptation mechanisms in the deployment of CLIP [7, 8]. However, a significant challenge remains: swiftly and

*Equal contribution.

Correspondence to david.osowiechi.1@ens.etsmtl.ca and mehrdad.noori.1@ens.etsmtl.ca

38th Conference on Neural Information Processing Systems (NeurIPS 2024).

| | Template |
|---|---|
| $T^0$: | “a photo of a {class $k$}” |
| $T^1$: | “itap of a {class $k$}” |
| $T^2$: | “a bad photo of the {class $k$}” |
| $T^3$: | “a origami {class $k$}” |
| $T^4$: | “a photo of the large {class $k$}” |
| $T^5$: | “a {class $k$} in a video game” |
| $T^6$: | “art of the {class $k$}” |
| $T^7$: | “a photo of the small {class $k$}” |

(a)

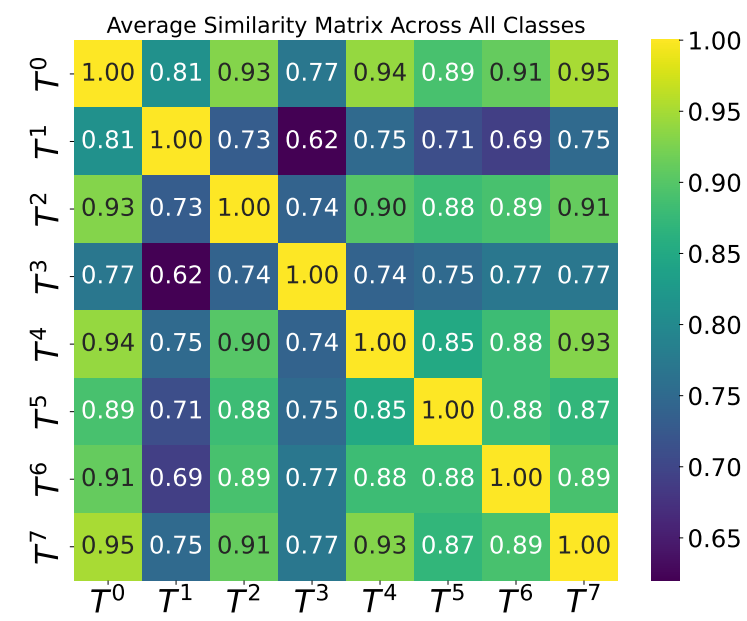

(b)

| | $T^0$ | $T^1$ | $T^2$ | $T^3$ | $T^4$ | $T^5$ | $T^6$ | $T^7$ | WATT |
|---|---|---|---|---|---|---|---|---|---|
| Original | 89.80 | 90.37 | 90.50 | 88.42 | 89.93 | 89.95 | 90.13 | 88.54 | **91.05** |
| Gaussian Noise | 60.19 | 61.01 | 61.17 | 58.24 | 58.84 | 58.35 | 59.62 | 61.13 | **63.84** |
| Defocus Blur | 77.23 | 77.07 | 78.00 | 75.98 | 76.39 | 77.45 | 77.08 | 75.59 | **78.94** |
| Snow | 76.57 | 77.36 | 77.93 | 75.08 | 77.45 | 77.09 | 77.05 | 75.57 | **79.79** |
| JPEG Compression | 64.65 | 65.36 | 65.24 | 64.16 | 64.18 | 64.36 | 64.78 | 65.32 | **67.36** |

(c)

Figure 1: (a) The different templates used during our experiments. (b) Matrix of cosine similarity between each template, averaged over all classes of the CIFAR-10 dataset. (c) Comparison of accuracy (%) using cross-entropy (CE) on CIFAR-10 and some corruptions of CIFAR-10-C datasets using different templates and our weight average strategy.

effectively adapting the model to new domains while preserving its attractive zero-shot capabilities, thus obviating the need for retraining.

To tackle this challenge, we investigate the impact of different text prompt templates, listed in Figure 1a, on model adaptation. Despite these templates being related, as reported in Figure 1b, the cosine similarity between their embeddings varies greatly, suggesting that they encode complementary information about the classes. The variability of information in different templates can also be observed in Figure 1c, where the classification accuracy obtained with these templates on a subset of CIFAR-10 corruptions fluctuates by up to 3%. Given this insight, finding an effective way to leverage the knowledge from different text templates would be useful to yield a better adaptation. This motivates our work proposing Weight Average adaptation during Test-Time (WATT).

By strategically averaging the adapted weights derived from multiple text prompt templates, our method aims to harness the complementary strengths of individual templates, resulting in robust and enhanced performance across a wide range of domain shifts. To further illustrate this point, Figure 2 presents the test loss and adaptation error surfaces for three models that are separately adapted using three templates ($T^0$, $T^1$, $T^2$) under the Gaussian noise corruption of the CIFAR-10-C dataset[2]. The central point in these landscapes, representing the final model obtained by averaging the weights of the separate models, demonstrates a convergence towards lower loss and error, highlighting the potential of weight averaging for test-time adaptation. Moreover, inspired by recent advancements in machine learning utilizing train-time weight averaging techniques [10, 11], the proposed WATT method can dynamically adjust to new data to tackle unforeseen distribution shifts without relying on class supervision.

We outline the main contributions of our work as follows:

- We introduce a novel Test-Time Adaptation method for CLIP, which leverages weight averaging across various text templates at test-time.

[2]To visualize the loss and error surface, we use weight vectors from models adapted with text templates $T^0$, $T^1$, and $T^2$, denoted as $w_0$, $w_1$, and $w_2$. Following [9], we define $u = w_1 - w_0$ and $v = (w_2 - w_0) - \frac{(w_2-w_0)\cdot(w_1-w_0)}{\|w_1-w_0\|^2}(w_1 - w_0)$. The normalized vectors $\hat{u} = \frac{u}{\|u\|}$ and $\hat{v} = \frac{v}{\|v\|}$ form an orthonormal basis in the plane of $w_0$, $w_1$, and $w_2$. We create a Cartesian grid in this basis and evaluate the networks at each grid point. A point $P$ with coordinates $(x, y)$ in the plane is given by $P = w_0 + x \cdot \hat{u} + y \cdot \hat{v}$. To plot all in the same plane, we used the average of the three templates' text embeddings.

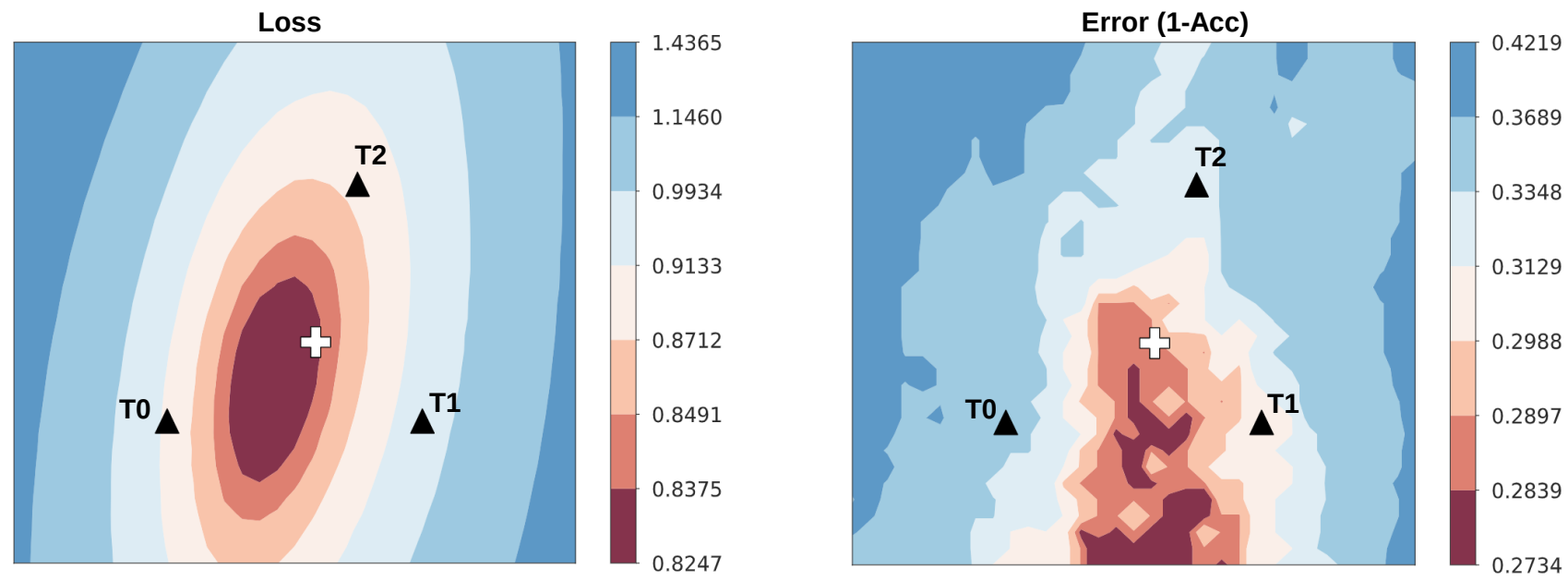

Figure 2: Loss and Error surfaces on model parameters for the Gaussian noise corruption of the CIFAR-10C dataset. Points $T^0$, $T^1$, and $T^2$ represent models adapted with different text templates (please see Fig. 1a). The central point (cross) shows the model obtained by averaging these weights, demonstrating improved performance.

- Our WATT strategy yields highly competitive performances in comparisons to current TTA methods, and could bring improvement using only a single image at test time, a capability not present in previous approaches.
- We rigorously evaluate our WATT methodology through comprehensive evaluations across different datasets characterized by diverse types and degrees of domain shifts, encompassing **a total of 155 evaluation scenarios**. Our experiments demonstrate the robustness and efficacy of WATT compared to existing adaptation methods.

## 2 Related work

**Test-Time Adaptation (TTA)** is crucial in domain adaptation, particularly with unlabeled target domain data and no access to source domain samples. The challenge lies in estimating the target domain's distribution and comparing domain characteristics indirectly. Recent advancements have highlighted the potential and limitations of adapting pre-trained models. A key focus has been on leveraging batch normalization layers for adaptation due to their ability to retain source domain information. Methods such as PTBN [12] and TENT [13] recalibrate batch statistics and optimize affine parameters via entropy minimization, though they often require image augmentations or large batches. MEMO [14] proposes a simple approach that does not require multiple test points; it uses single-test-point data augmentations and minimizes the marginal entropy of the model's average output to enforce consistency and improve robustness. LAME [15] introduces a closed-form optimization strategy that refines model predictions for target images by leveraging the Laplacian of feature maps to encourage clustering, thereby emphasizing feature similarities. In another study, SAR [16] addresses TTA instability by using batch-agnostic norms (e.g., group and layer norms) and a sharpness-aware entropy minimization approach. It filters noisy samples and guides the model to flat minima, improving robustness under mixed domain shifts and small batch sizes.

Recently, Test-Time Training (TTT) methodologies have emerged as prominent contenders in TTA [17, 18, 19, 20, 21]. This approach involves training a supplementary sub-branch alongside the primary network in an unsupervised manner, subsequently leveraging it to refine the model. Unlike previous methods, our approach operates on individual image batches, offering a significant advantage in TTA by avoiding the necessity of training additional branches from scratch.

In natural language processing, TPT [8] introduced entropy minimization for adapting models like CLIP, albeit with high computational costs due to learning an adapter at the text prompt with multiple transformations. DiffTPT [22] extends this by leveraging pre-trained diffusion models to generate diverse and informative augmented data, combining conventional augmentation methods used in TPT with diffusion-generated data to enhance adaptability. It also introduces a cosine similarity-based filtration technique for improved prediction fidelity. TDA [23] offers a training-free approach with a dynamic adapter, utilizing a lightweight key-value cache and pseudo label refinement, making it computationally efficient. CLIPArTT [24], fine-tunes normalization layers with minimal disruption to the model's knowledge, enhancing text supervision by introducing pseudo labels. Existing methods often lag behind supervised prompt adaptation techniques in performance. SwapPrompt [25] bridges this gap by leveraging self-supervised contrastive learning, employing a dual prompt paradigm. In

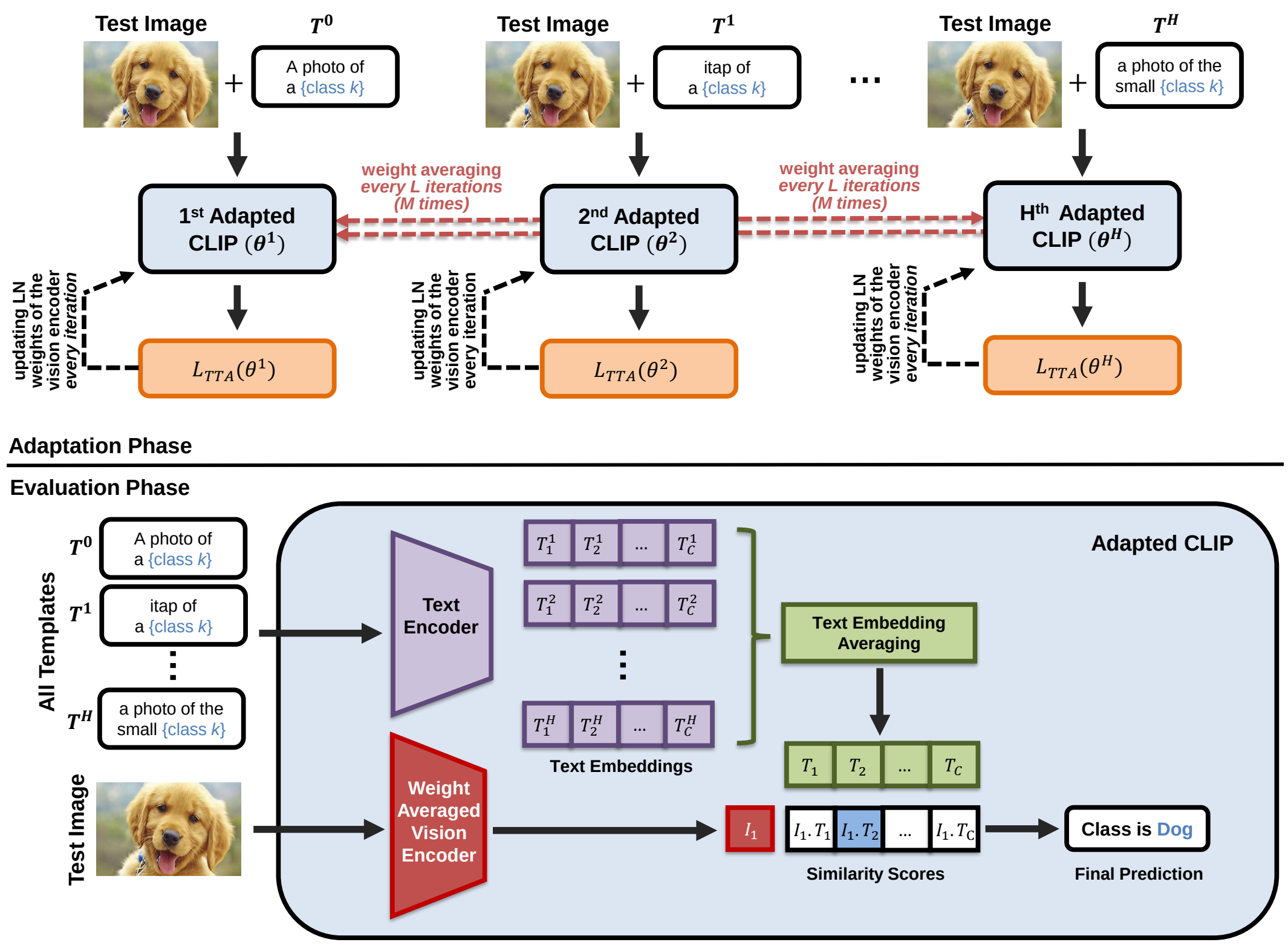

Figure 3: Overview of the proposed WATT method. In the Adaptation Phase, the model is adapted using different text templates ($T^0$, $T^1$, ..., $T^H$), with weight averaging performed periodically. In the Evaluation Phase, the adapted CLIP model uses averaged text embeddings from all templates and the weight averaged model to predict the class of the test image.

comparison, our method combines prompt augmentation and fine-tuning of normalization layers, highlighting its effectiveness in test-time adaptation.

**Weight Averaging (WA)** is a powerful train-time technique for improving deep neural network generalization. Stochastic Weight Averaging (SWA) [26] averages weights of multiple models sampled from different training epochs, aiding smooth optimization trajectory and convergence to points with superior generalization. SWAD [10] refines SWA by densely sampling weights throughout training, enhancing generalization and robustness across tasks. This train-time refinement enhances WA's effectiveness in producing models with improved generalization. The Lookaround [11] optimizer iterates between an "around step" and an "average step", building on SWAD's advancements. In the "around step", independently trained models using various data augmentations explore a broader loss landscape to find flatter minima. In the "average step," weights of these models are then averaged, guiding optimization towards lower loss regions. This method enhances robustness and generalization across tasks, improving upon SWA and SWAD by providing a more effective weight averaging process.

In contrast to existing approaches, WATT leverages varied text prompts to adapt vision-language models such as CLIP during testing. Our method also harnesses the benefits of weight averaging while addressing domain shifts without additional model transformations or trainable modules, thereby setting a new precedent in test-time adaptation.

## 3 Method

The proposed WATT method, summarized visually in Figure 3 comprises three main components, the first two in the Adaptation Phase and the third in the Evaluation Phase: **1)** a light-weight transductive TTA strategy that adapts CLIP's visual encoder effectively by considering the similarity between *all*

| Dataset | single_temp | text_avg |
|---|---|---|
| CIFAR-10 | 90.87 ±0.10 | **91.08** **±0.06** |
| CIFAR-10.1 | 86.80 ±0.19 | **86.85** **±0.18** |
| CIFAR-10-C | 72.08 | **72.66** |
| CIFAR-100 | 69.79 ±0.20 | **70.30** **±0.11** |
| CIFAR-100-C | 41.79 | **42.24** |

Table 1: Accuracy (%) with different text ensembles at test time.

batch samples in terms of their visual and text features; **2)** a weight-averaging strategy using multiple text templates to generate diverse model hypotheses during adaptation; **3)** an ensembling technique that boosts performance during evaluation by averaging the embedding of different text templates.

### 3.1 Transductive TTA loss

While our method can be employed with any TTA framework, in this work, we implement a strategy inspired by the transductive TTA approach of CLIPArTT [24] which effectively incorporates semantic relationships among batch samples.

Initially, our process involves executing inference using CLIP, a system comprising a visual encoder $f^v_\theta(\cdot)$ that translates an image $\mathbf{x}$ into visual features $\mathbf{z}^v \in \mathbb{R}^D$, and a text encoder $f^t_\theta(\cdot)$ which converts text prompts $\mathbf{t}$ into text features $\mathbf{z}^t \in \mathbb{R}^D$. During inference, we employ pre-defined text prompts assigned to each class within a dataset, such as $\mathbf{t}^0_k$ = "`a photo of a {class` $k$`}`". For a new image $\mathbf{x}_i$, the likelihood of belonging to class $k$ is then computed using cosine similarity:

$$p_{ik} = \frac{\exp\big(\cos(\mathbf{z}^v_i, \mathbf{z}^t_k)/\tau\big)}{\sum_j \exp\big(\cos(\mathbf{z}^v_i, \mathbf{z}^t_j)/\tau\big)}, \quad \cos(\mathbf{z}, \mathbf{z}') = \frac{\mathbf{z}^\top \mathbf{z}'}{\|\mathbf{z}\|_2 \cdot \|\mathbf{z}'\|_2}, \tag{1}$$

where $\tau$ is a softmax temperature parameter set to $0.01$ is this work. This prediction is then stored to be used as pseudo labels for the model.

Denoting the normalized visual embeddings of the samples within the test batch as $\mathbf{Z}^v \in \mathbb{R}^{B\times D}$ and the instance-specific text embeddings as $\mathbf{Z}^t \in \mathbb{R}^{B\times D}$, we compute an image-to-image similarity matrix $\mathbf{S}^v = \mathbf{Z}^v(\mathbf{Z}^v)^\top \in [-1,1]^{B\times B}$ modeling pairwise relationships in terms of image characteristics. Similarly, we construct a text-to-text similarity matrix $\mathbf{S}^t = \mathbf{Z}^t(\mathbf{Z}^t)^\top \in [-1,1]^{B\times B}$, capturing the semantic relationships among text embeddings within the batch. Utilizing the computed pairwise similarity matrices, we generate pseudo labels $\mathbf{Q} = \mathrm{softmax}\big((\mathbf{S}^v + \mathbf{S}^t)/2\tau\big) \in [0,1]^{B\times B}$ which are used with cross-entropy in our transductive TTA loss:

$$\mathcal{L}_{\mathrm{TTA}}(\theta) = -\frac{1}{B}\sum_{i=1}^{B}\sum_{j=1}^{B} q_{ij} \log p_{ij}. \tag{2}$$

Drawing a link with the Stochastic Neighbor Embedding (SNE) method for dimensionality reduction [27], which minimizes the KL divergence between distributions modeling pairwise distances, our TTA loss ensures that the inter-modality (text-to-image) similarities of batch samples are aligned with their intra-modality ones (text-to-text and image-to-image).

### 3.2 Multi-Template Weight Averaging

We explore various text prompt templates suggested in the CLIP paper and detailed in Fig. 1a. As reported in Table 1c, these prompts achieve varying performance across different corruption types of CIFAR-10-C. We formulate prompts of the form $\mathbf{t}^h_k$ = `template` $h$`(class` $k$`)`, where $h \in \{1, 2, \ldots, H\}$, encompassing a spectrum of textual cues tailored to elicit diverse responses from the model.

Two different approaches are investigated for our multi-template weight averaging (MTWA) strategy. The first one denoted as **Parallel MTWA (WATT-P)**, which follows recent optimization approaches like Lookaround [11], performs the adaptation separately for each text template, starting from the same parameters, and then averages the resulting adapted weights. The second one, called **Sequential MTWA (WATT-S)**, instead considers text templates sequentially without resetting the weights. These two approaches, which we illustrate and compare in Fig. 4, are detailed below.

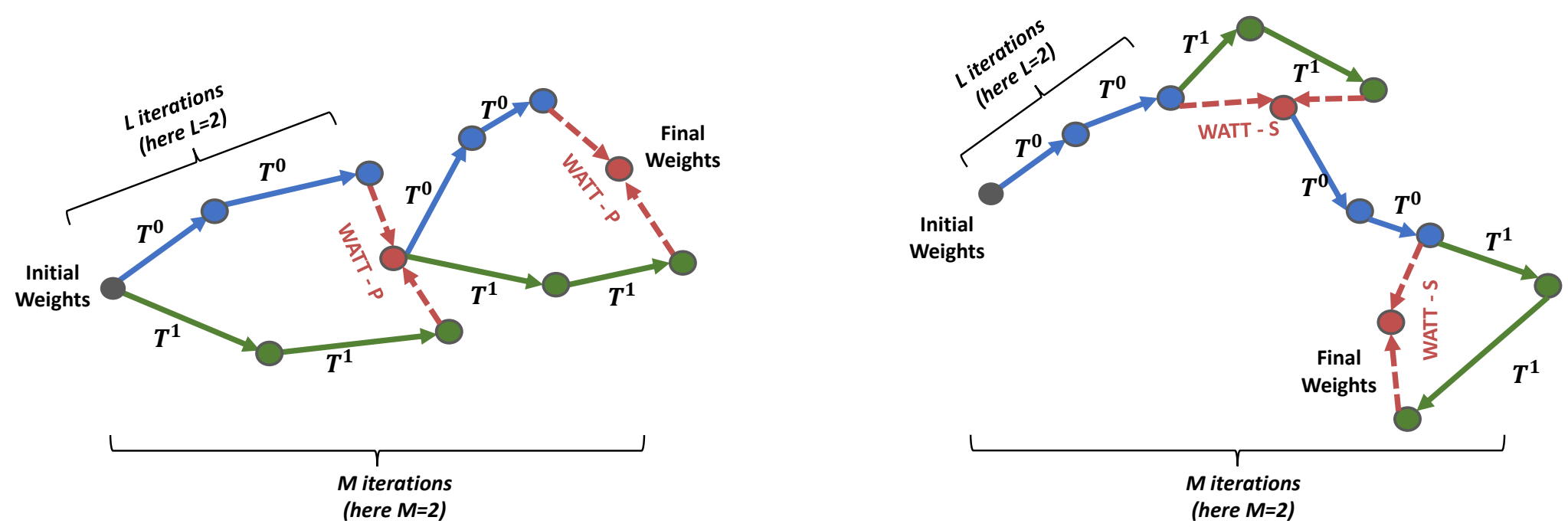

Figure 4: Visual comparison of the Parallel **(left)** and Sequential **(right)** approaches for multi-template weight averaging during adaptation.

| Dataset | CLIP | BS = 1 | BS = 2 | BS = 4 | BS = 8 | BS = 16 | BS = 32 | BS = 64 | BS = 128 |
|---|---|---|---|---|---|---|---|---|---|
| CIFAR-10 | 88.74 | 89.87 | 89.39 ±0.02 | 89.16 ±0.07 | 88.93 ±0.16 | 89.14 ±0.04 | 89.51 ±0.12 | 90.16 ±0.13 | 91.05 ±0.06 |
| CIFAR-10.1 | 83.25 | 84.55 | 84.32 ±0.15 | 83.88 ±0.17 | 84.12 ±0.37 | 84.35 ±0.21 | 84.87 ±0.16 | 85.52 ±0.30 | 86.98 ±0.31 |
| CIFAR-10-C | 59.22 | 61.26 | 63.60 | 63.47 | 63.94 | 65.66 | 68.34 | 71.21 | 73.82 |

Table 2: Accuracy (%) of our method for different batch sizes compared to CLIP.

**Parallel MTWA.** This approach optimizes the TTA loss in (2) separately for $H$ different models, each utilizing a distinct template. Starting from the same visual encoder parameters $\theta$, these models are updated in parallel for $L$ iterations, resulting in updated parameters $\theta'_h$, with $h \in \{1, \ldots, H\}$. The parameters are reset after each update, enabling each model to restart the adaptation from the same initial point. Subsequently, we aggregate the weights obtained from these $H$ models by computing their average: $\theta_{\text{avg}} = \frac{1}{H} \sum_{h=1}^{H} \theta'_h$. We repeat this step $M$ times, and denote the overall process as "(after $L$ iter) $\times$ $M$".

**Sequential MTWA.** Our Sequential MTWA approach is inspired from the work of [26], where the averaging of weights across various stages of the training process is employed to mitigate variance and enhance generalization capabilities. Instead of resetting parameters for each model, we update parameters after each template's iteration. To ensure impartiality in the update sequence of templates, a random selection process is implemented, thereby disregarding any predetermined order.

### 3.3 Evaluation Phase

At test-time, predictions are computed using Equation 1 through two distinct methodologies. In the first approach, the text features $\mathbf{z}_0^t$ are derived from the initial text prompt $\mathbf{t}_k^0 =$ "`a photo of a class` $k$", denoted as single_temp. Conversely, the second method aggregates the text features from all templates by computing their mean, resulting in the prediction $\mathbf{z}_{\text{ens}}^t = \frac{1}{H} \sum_{h=1}^{H} \mathbf{z}_h^t$, denoted as text_avg (see Fig. 3).

## 4 Experimental Setup

**Settings.** In line with prior TTA methodologies, adjustments are made to all Layer Normalization layers within the visual feature extractor for test-time adaptation. The Adam optimizer is employed with a fixed learning rate of $10^{-3}$, wheras a smaller learning rate of $10^{-4}$ is chosen for adaptation to the 3D renderings split, as it reflects a more pronounced shift. Throughout our experimentation process, a consistent batch size of 128 is maintained to ensure uniformity and facilitate meaningful comparisons across various scenarios.

**Datasets.** Following [24], we rigorously evaluate WATT's performance across diverse TTA datasets using established assessment techniques. These datasets simulate intricate domain shifts, providing nuanced insights into our approach's effectiveness. Additionally, we explore WATT's adaptability on the original dataset through zero-shot test-time adaptation. To ensure a thorough examination, we

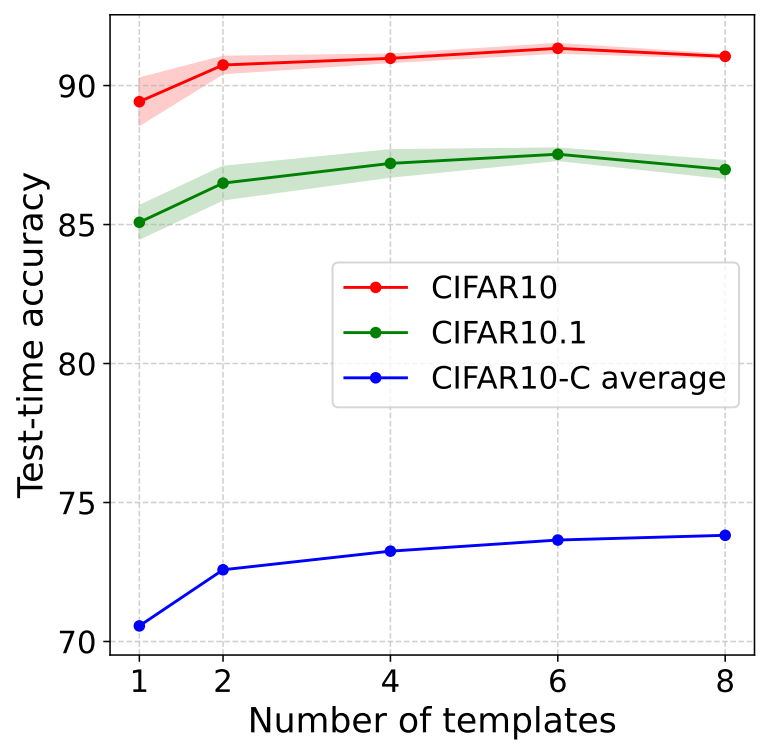

Figure 5: Evolution of the accuracy for different numbers of random template on 5 test-time runs.

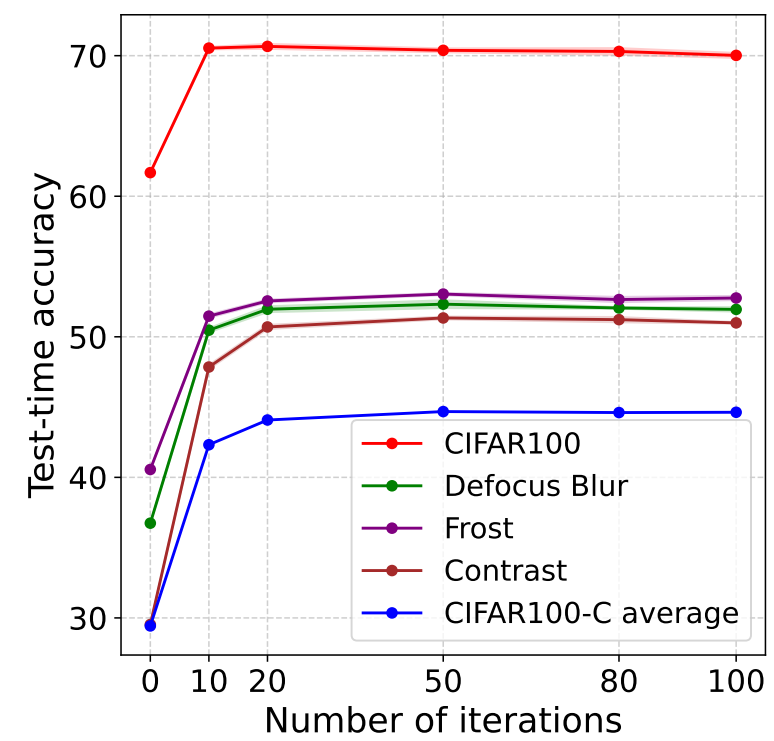

Figure 6: Evolution of accuracy on CIFAR-100 corruptions with the Parallel MTWA method.

extend our analysis to include various domain generalization datasets, exposing our method to a broad spectrum of image categories for comprehensive evaluation. Our evaluation framework encompasses *natural images*, *common corruptions*, *simulated shifts*, *video shifts*, *texture shifts* and *style shifts*.

In our assessment of natural image analysis, we include CIFAR-10, CIFAR-10.1, and CIFAR-100, each comprising 10,000 images and offering varied data distributions. CIFAR-10.1 [28] introduces a natural shift from CIFAR-10, providing a comprehensive evaluation of our model's performance. We also incorporate the CIFAR-10-C and CIFAR-100-C datasets [28], augmented with 15 distinct corruptions across 5 severity levels, resulting in 75 common corruption scenarios. This comprehensive augmentation assesses the model's resilience effectively.

Our investigation also extends to the VisDA-C dataset [29], challenging models with simulated and video shifts across diverse imagery types. Additionally, we evaluate our method on three datasets mostly used in the field of domain generalization: PACS [30], VLCS [31], and OfficeHome [32] datasets, instrumental in understanding texture and style variations. These evaluations effectively demonstrate the generalizability of our method across distinct domain shifts.

**Benchmarking.** We conduct a comparative analysis of WATT against contemporary methods using ViT-B/32 as the backbone. Specifically, we incorporate an adapted version of TENT [13], customized for CLIP by the authors of [24], using 10 iterations. We also include SAR [16] and MEMO [14] in the same manner as TENT. Additionally, we compare with TPT [8], a novel adaptation technique for CLIP that heavily relies on image augmentations, as well as DiffTPT [22], an extension that incorporates diffusion models. Lastly, we include TDA [23], which employs a dynamic adapter, and CLIPArTT [24], a recent approach that utilizes pseudo labels generated through conformal learning.

## 5 Results

In this section, we present empirical findings from our WATT method through a series of ablation studies aimed at understanding the impacts of individual components. These studies inform subsequent experiments across diverse datasets. Leveraging insights from ablations, we conduct comprehensive experiments, benchmarking WATT against state-of-the-art techniques across various datasets.

### 5.1 Ablation Studies

In this section, unless otherwise specified, we focus on the Sequential MTWA variant of our method (see Section 3.2) and will use these findings as a reference for the Parallel MTWA method.

**Comparison of the template used during testing.** After updating the model, we proceed to compute the similarity between the image features and the text embeddings, enabling prediction. Typically, text embeddings originate from the text prompt "`a photo of a class` $k$". However, by employing multiple templates, we have the flexibility to alter this text embedding through the averaging of all text embeddings from each template. Table 1 conducts a comparative analysis revealing that this

averaged text embedding consistently yields superior results across all scenarios. Hence, we adopt this approach for next experiments.

**Comparison of the number of templates.** In Figure 5, we examine the performance variation relative to the number of utilized templates. In this investigation, we conduct 5 runs wherein the templates are randomly selected from a pool of 8 distinct templates (as outlined in Fig. 1a). Notably, when the distribution shift is minimal, as observed in CIFAR-10 and CIFAR-10.1, optimal performance is attained using 6 templates, with performance gradually diminishing thereafter. Conversely, in scenarios characterized by substantial corruptions, such as CIFAR-10-C, employing all 8 templates proves advantageous. Consequently, our focus moving forward will be on utilizing all 8 templates in our work.

| Dataset | Text avg. | Output avg. | Weight avg. (ours) | | |
|---|---|---|---|---|---|
| | | | (after 10 iter)×1 | (after 1 iter)×10 | (after 2 iter)×5 |
| CIFAR-10 | 90.58 ±0.03 | 90.90 ±0.03 | 91.08 ±0.06 | **91.39 ±0.14** | 91.05 ±0.06 |
| CIFAR-10.1 | 85.78 ±0.25 | 86.77 ±0.08 | 86.85 ±0.18 | **88.02 ±0.18** | 86.98 ±0.31 |
| CIFAR-10-C | 71.41 | 72.60 | 72.66 | 73.66 | **73.82** |
| CIFAR-100 | 69.46 ±0.13 | 70.32 ±0.1 | 70.3 ±0.11 | **70.85 ±0.08** | 70.74 ±0.20 |
| CIFAR-100-C | 41.37 | 42.68 | 42.24 | 45.32 | **45.57** |

Table 3: Accuracy (%) obtained with different averaging strategies.

**Text Averaging vs Output Averaging vs Weight Averaging.** Utilizing the averaging method within a VLM offers several possibilities, including averaging the weights, the outputs or the text embeddings before computing the logits. In Table 3, a comparison between these approaches is presented. It becomes evident that weight averaging consistently outperforms text embedding averaging across various datasets, showcasing a superiority of approximately 1% even with the less effective weight averaging method. This performance advantage is observed across CIFAR-10, CIFAR-10.1, and CIFAR-10-C, and persists even with larger numbers of classes, such as in CIFAR-100 and CIFAR-100-C. When concentrating on output averaging, the results may be less evident with less effective weight averaging methods. However, they remain valid and even more accurate with superior weight averaging techniques. Therefore, our focus for future experiments will be on weight averaging as the preferred approach.

**Best moment to do the Weight Averaging.** Examining Table 3, it is evident that the parameters $L$ and $M$ discussed in Section 3 are crucial. Specifically, a large $L$ (e.g., 10) combined with a small $M$ (e.g., 1) is ineffective. Conversely, setting $L = 1$ and $M = 10$ yields optimal results for small distribution shift datasets, while $L = 2$ and $M = 5$ perform best on highly corrupted datasets. Given that TTA typically encounters substantial distribution shifts, we will use $L = 2$ and $M = 5$ in our subsequent experiments.

**Performance over the number of iterations.** In this section, we focus on the method incorporating a Parallel MTWA mechanism and examine the impact of the number of iterations on performance. As illustrated in Figure 6, the accuracy stabilizes after approximately 20 iterations. Although there is a slight improvement in performance beyond 50 iterations, the difference is marginal. Based on these observations, we have opted to use 50 iterations for our experiments.

**Model performance across various batch sizes.** In our investigation, we delve into the performance implications of TTA methods when operating under small batch sizes, a historical challenge in the field. Table 2 provides insights into this aspect, revealing substantial performance enhancements with increasing batch sizes. Notably, our WATT model showcases remarkable adaptability, demonstrating performance improvements even with a single image input contrary to alternative methods. Specifically, we observe enhancements of approximately 1% for CIFAR-10 and CIFAR-10.1, and an impressive 2% for CIFAR-10-C when compared to baseline. Moving forward, we maintain a batch size of 128 in our experiments, aligning with prevalent practices observed in contemporary state-of-the-art methodologies.

| Dataset | | CLIP | TENT | TPT | TDA | DiffTPT | SAR | CLIPArTT | WATT-P | WATT-S |
|---|---|---|---|---|---|---|---|---|---|---|
| CIFAR-10 | | 88.74 | **91.69** ±0.10 | 88.06 ±0.06 | 84.09 ±0.04 | 83.07 ±0.05 | 89.05 ±0.02 | 90.04 ±0.13 | 91.41 ±0.17 | 91.05 ±0.06 |
| CIFAR-10.1 | | 83.25 | 87.60 ±0.45 | 81.80 ±0.27 | 78.98 ±0.37 | 76.50 ±0.29 | 83.65 ±0.04 | 86.35 ±0.27 | **87.78 ±0.05** | 86.98 ±0.31 |
| CIFAR-10-C | | 59.22 | 67.56 | 56.80 | 48.00 | 56.77 | 60.45 | 71.17 | 72.83 | **73.82** |
| CIFAR-100 | | 61.68 | 69.74 ±0.16 | 63.78 ±0.28 | 60.32 ±0.06 | 52.80 ±0.08 | 64.44 ±0.01 | 69.79 ±0.04 | 70.38 ±0.14 | **70.74 ±0.20** |
| CIFAR-100-C | Gaussian Noise | 14.80 | 14.38 ±0.14 | 14.03 ±0.10 | 8.20 ±0.35 | 21.40 ±0.08 | 15.85 ±0.03 | 25.32 ±0.14 | 31.28 ±0.03 | **32.07 ±0.23** |
| | Shot noise | 16.03 | 17.34 ±0.27 | 15.25 ±0.17 | 9.58 ±0.43 | 24.17 ±0.49 | 17.41 ±0.05 | 27.90 ±0.05 | 33.44 ±0.11 | **34.36 ±0.11** |
| | Impulse Noise | 13.85 | 10.03 ±0.13 | 13.01 ±0.13 | 7.63 ±0.19 | 16.87 ±0.24 | 14.90 ±0.09 | 25.62 ±0.09 | 29.40 ±0.11 | **30.33 ±0.03** |
| | Defocus blur | 36.74 | 49.05 ±0.07 | 37.60 ±0.17 | 25.59 ±0.41 | 20.30 ±0.29 | 42.00 ±0.06 | 49.88 ±0.23 | 52.32 ±0.28 | **52.99 ±0.16** |
| | Glass blur | 14.19 | 3.71 ±0.07 | 16.41 ±0.02 | 9.83 ±0.56 | 15.57 ±0.46 | 13.84 ±0.08 | 27.89 ±0.03 | 31.20 ±0.12 | **32.15 ±0.30** |
| | Motion blur | 36.14 | 46.62 ±0.27 | 37.52 ±0.23 | 28.92 ±0.18 | 21.00 ±0.64 | 39.52 ±0.01 | 47.93 ±0.14 | 49.72 ±0.15 | **50.53 ±0.12** |
| | Zoom blur | 40.24 | 51.84 ±0.15 | 42.99 ±0.11 | 31.08 ±0.36 | 25.53 ±0.05 | 45.40 ±0.05 | 52.70 ±0.06 | 54.72 ±0.04 | **55.30 ±0.22** |
| | Snow | 38.95 | 46.71 ±0.21 | 42.35 ±0.13 | 32.94 ±0.12 | 28.83 ±0.37 | 41.85 ±0.08 | 49.72 ±0.01 | 51.79 ±0.04 | **52.77 ±0.15** |
| | Frost | 40.56 | 44.90 ±0.27 | 43.31 ±0.14 | 34.84 ±0.25 | 31.10 ±0.36 | 42.20 ±0.04 | 49.63 ±0.12 | 53.04 ±0.08 | **53.79 ±0.31** |
| | Fog | 38.00 | 47.31 ±0.04 | 38.81 ±0.17 | 31.13 ±0.15 | 16.60 ±0.43 | 40.14 ±0.00 | 48.77 ±0.04 | 50.78 ±0.24 | **51.49 ±0.21** |
| | Brightness | 48.18 | 60.58 ±0.18 | 50.23 ±0.11 | 42.36 ±0.10 | 38.13 ±0.29 | 52.77 ±0.10 | 61.27 ±0.08 | 62.65 ±0.25 | **63.57 ±0.21** |
| | Contrast | 29.53 | 45.90 ±0.11 | 28.09 ±0.09 | 18.03 ±0.07 | 7.70 ±0.22 | 34.40 ±0.10 | 48.55 ±0.24 | 51.34 ±0.10 | **52.76 ±0.27** |
| | Elastic transform | 26.33 | 33.09 ±0.08 | 28.12 ±0.15 | 18.88 ±0.24 | 21.60 ±0.51 | 28.44 ±0.07 | 37.45 ±0.08 | 39.97 ±0.06 | **40.90 ±0.43** |
| | Pixelate | 21.98 | 26.47 ±0.09 | 20.43 ±0.14 | 14.59 ±0.30 | 22.83 ±0.31 | 22.91 ±0.07 | 33.88 ±0.14 | 39.59 ±0.09 | **40.97 ±0.16** |
| | JPEG compression | 25.91 | 29.89 ±0.07 | 28.82 ±0.09 | 17.56 ±0.11 | 31.77 ±0.45 | 27.20 ±0.06 | 36.07 ±0.32 | 38.99 ±0.16 | **39.59 ±0.08** |
| | Mean | 29.43 | 35.19 | 30.46 | 22.08 | 22.89 | 31.92 | 41.51 | 44.68 | **45.57** |

Table 4: Accuracy (%) on CIFAR-10, CIFAR-10.1, CIFAR-10-C, CIFAR-100, and CIFAR-100-C datasets. WATT-P refers to our method with Parallel MTWA and WATT-S to the Sequential MTWA variant of WATT.

### 5.2 Comparison to SOTA methods

**Performance evaluation with small batches.** In existing TTA works that study small batch size scenarios, such as SAR [16] and MEMO [14], we implement these methods in the context of CLIP. We compare our method with other TTA approaches using a batch size of 1. As shown in Table 5, WATT-P achieves the highest accuracy across all cases, outperforming SAR by 2.23% and MEMO by 0.75% on CIFAR-10.1. Notably, this improvement is achieved without any image augmentation, which is a common practice in previous TTA approaches that deal with small batches.

| Dataset | CLIP | TPT | SAR | MEMO | CLIPArTT | WATT-P |
|---|---|---|---|---|---|---|
| CIFAR-10 | 88.74 | 88.29 | 87.41 | 89.29 | 88.76 | 89.87 |
| CIFAR 10.1 | 83.25 | 82.85 | 82.32 | 83.80 | 83.15 | 84.55 |
| CIFAR-10-C | 59.22 | 59.03 | 58.70 | 61.15 | 59.18 | 61.26 |

Table 5: Comparison of different TTA methods with a batch size equal to 1.

**Performance evaluation in the presence of natural or no domain shift.** In Table 4, results show consistent performance enhancements with WATT, both with the Parallel and Sequential MTWA strategies, alongside the baseline. On CIFAR-10, performance improves by 2.67% with Parallel MTWA and 2.31% using Sequential MTWA. On CIFAR-10.1, improvements reach 4.53% and 3.73%, and on CIFAR-100, enhancements are 8.70% and 9.06%. While WATT consistently outperforms the baseline, TPT, and CLIPArTT, TENT yields superior results on CIFAR-10. WATT's effectiveness often correlates with the number of classes, showing better performance with more classes, indicating its strength in lower-confidence scenarios.

**Performance evaluation in the presence of common corruptions.** Table 4 shows that both WATT variants consistently outperform alternative methods across various corruptions and class numbers. Notably, WATT with Parallel MTWA improves performance by 16.48% on CIFAR-100 *Gaussian Noise* and by 17.01% on *Glass Blur* compared to the baseline, while WATT with Sequential MTWA shows improvements of 17.27% and 17.96% respectively. On common corruptions, the Sequential MTWA variant surpasses Parallel MTWA, with improvements of 0.99% on CIFAR-10 and 0.89% on CIFAR-100. According to the TDA supplementary materials, we selected the weighting factor alpha as 5.0 and the sharpness ratio beta as 2.0, which are stated as optimal. However, these values did not appear to be the best choice for more challenging datasets like CIFAR-10-C or CIFAR-100-C that contain various corruptions. Adjusting these parameters based on the dataset would not be consistent

| Dataset | Domain | CLIP | TENT | TPT | CLIPArTT | WATT-P | WATT-S |
|---|---|---|---|---|---|---|---|
| VisDA-C | 3D (trainset) | 84.43 | 84.86 ±0.01 | 79.35 ±0.04 | 85.09 ±0.01 | **85.42 ±0.03** | 85.36 ±0.01 |
| | YT (valset) | 84.45 | 84.68 ±0.01 | 83.57 ±0.04 | 84.40 ±0.01 | 84.57 ±0.00 | **84.69 ±0.01** |
| | Mean | 84.44 | 84.77 | 81.46 | 84.75 | 85.00 | **85.03** |
| OfficeHome | Art | 73.75 | 74.03 ±0.27 | **75.76 ±0.27** | 73.84 ±0.20 | 75.65 ±0.27 | **75.76 ±0.39** |
| | Clipart | 63.33 | 63.42 ±0.04 | 63.08 ±0.31 | 63.54 ±0.06 | **66.23 ±0.13** | 65.77 ±0.11 |
| | Product | 85.32 | **85.51 ±0.08** | 84.07 ±0.28 | 85.23 ±0.16 | 85.41 ±0.09 | 85.41 ±0.01 |
| | Real World | 87.71 | 87.74 ±0.05 | 85.89 ±0.33 | 87.61 ±0.05 | 88.22 ±0.15 | **88.37 ±0.05** |
| | Mean | 77.53 | 77.68 | 77.20 | 77.56 | **78.88** | 78.83 |
| PACS | Art | 96.34 | **96.65 ±0.05** | 95.52 ±0.20 | 96.57 ±0.09 | 96.31 ±0.00 | 96.39 ±0.00 |
| | Cartoon | 96.08 | 96.22 ±0.05 | 94.77 ±0.20 | 96.00 ±0.02 | 96.52 ±0.02 | **96.62 ±0.02** |
| | Photo | 99.34 | 99.40 ±0.00 | 99.42 ±0.06 | 99.28 ±0.00 | 99.48 ±0.03 | **99.52 ±0.00** |
| | Sketch | 82.85 | 82.96 ±0.12 | 83.22 ±0.14 | 83.93 ±0.14 | **86.92 ±0.04** | 86.65 ±0.12 |
| | Mean | 93.65 | 93.81 | 93.23 | 93.95 | **94.81** | 94.80 |
| VLCS | Caltech101 | **99.51** | **99.51 ±0.00** | 99.36 ±0.06 | **99.51 ±0.00** | 99.43 ±0.00 | **99.51 ±0.00** |
| | LabelMe | 68.15 | 67.89 ±0.13 | 54.88 ±0.12 | 67.96 ±0.04 | 66.67 ±0.21 | **68.49 ±0.12** |
| | SUN09 | 68.85 | 69.27 ±0.04 | 67.30 ±0.49 | 68.68 ±0.09 | 72.61 ±0.15 | **73.13 ±0.17** |
| | VOC2007 | 84.13 | **84.42 ±0.15** | 76.74 ±0.28 | 84.09 ±0.02 | 82.30 ±0.16 | 83.41 ±0.17 |
| | Mean | 80.16 | 80.27 | 74.57 | 80.06 | 80.25 | **81.14** |

Table 6: Accuracy (%) on different domains of VisDA-C, OfficeHome, PACS and VLCS datasets.

with the principles of a fully TTA method, which might explain their suboptimal performance in our results. Regarding DiffTPT, it generates 64 images per test image, making it challenging to use in a real-world TTA scenario. Similar to TDA, DiffTPT requires carefully chosen parameters to fit the dataset, whereas our method does not require dataset-specific tuning. This highlights the robustness and practicality of our approach in diverse real-world applications.

**Performance analysis under simulated and video shifts.** Results on the 3D (simulated shift) and YT (video shift) splits of VisDA-C demonstrate a significant improvement in accuracy with our proposed WATT method compared to pure CLIP. The Sequential MTWA variant achieves the highest accuracy on both the 3D and YT splits, with scores of 85.36% and 84.69%, respectively, surpassing other adaptation methods including TENT, TPT, and CLIPArTT (see Table 6).

**Performance analysis under texture and style shifts.** Results on the OfficeHome, PACS, and VLCS datasets are presented in Table 6. On average, our proposed WATT method, with Parallel and Sequential MTWA variants, improves performance across the different domains of OfficeHome, PACS, and VLCS compared to other methods. This highlights its robustness in addressing texture and style shifts, which are especially challenging compared to other domain shift variants.

## 6 Conclusion

We introduce WATT, a Test-Time Adaptation method tailored for Vision-Language Models. Our approach harnesses Weight Averaging with different text prompts and incorporates text embeddings averaging to bolster prediction accuracy.

Through an extensive ablation study, we scrutinized the efficacy of employing varied text prompts and weight averaging. Comparative evaluations across Test-Time Adaptation and Domain Generalization datasets underscored the superiority of our method, particularly in scenarios involving distribution shifts and zero-shot performance enhancements compared to state-of-the-art approaches.

Looking forward, investigating the potential of text prompts and weight averaging in classification opens up promising avenues for future exploration. Our methodology, with its focus on template manipulation, suggests potential avenues for extension, such as incorporating alternative class descriptors, yielding valuable insights for future research. Moreover, expanding Test-Time Adaptation to encompass diverse scenarios, including segmentation or object detection with Vision-Language Models, holds significant potential for advancing our comprehension of model adaptability and performance across varied tasks.

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

# WATT: Weight Average Test-Time Adaption of CLIP
# Supplementary Material

## A Implementation

Our proposed WATT method is implemented in Python using the PyTorch (version 2.0.1) framework. All experiments were conducted on an NVIDIA V100 32 GB GPU. However, due to the effectively and lightweight nature of our method, it can be executed on less powerful GPUs. Specifically, the adaptation with a batch size of 128 using the ViT/B32 backbone requires up to 4 GB of memory, making it feasible to use on a wide range of GPUs. For fairness and consistency, we re-implemented and ran all other methods, including CLIPArTT, TENT, and TPT, in the same environment. Each experiment was performed three times to ensure reliability (three trials per experiment). To facilitate reproducibility, we provide the original implementation and detailed step-by-step instructions in our repository, accessible via *this github link*.

## B Template details

The CLIP paper identifies 80 templates that enhance model robustness and performance. They ultimately conclude that 7 of these templates best summarize their model (see *this link*). In our work, we use these 7 generic templates and add the common one, "`a photo of {}`," based on their optimization. These templates are not specifically linked to the content of the images.

## C Dataset details

In our investigation, we use the VisDA-C dataset, which challenges models with two distinct domain shifts: the simulated shift and the video shift. The simulated shift includes 152,397 3D-rendered images across 12 diverse classes, while the video shift comprises 72,372 YouTube video frames spanning the same categories. This dataset addresses the diversity of imagery types applicable to a model, posing a significant challenge.

Moreover, we evaluate our proposed method on three other datasets: PACS, VLCS, and OfficeHome. These datasets help understand various domain shifts, including texture and style variations. The PACS dataset consists of 9,991 images across four domains (Art, Cartoons, Photos, Sketches) and seven classes. The VLCS dataset contains 10,729 images across four domains (Caltech101, LabelMe, SUN09, VOC2007) and five classes. Lastly, the OfficeHome dataset includes 15,588 images across four domains (Art, Clipart, Product, Real) and 65 classes. Evaluating across these distinct domain shifts showcases the generalizability of our method. These datasets are more representative of real-world scenarios compared to CIFAR, with complex domain shifts.

## D Computational Cost

We conduct a thorough evaluation under consistent conditions using an NVIDIA A6000 GPU within the same Python environment. The Table 7 provided compares the adaptation time, memory usage, and the number of learnable parameters for various TTA methods, including our proposed WATT method.The table clearly demonstrates that WATT-S, a sequential implementation of WATT, maintains competitive adaptation time and memory usage compared to other methods like TENT and ClipArTT, which are efficient but lack the robustness of WATT's method. Additionally, the table highlights that WATT-P, with parallel model training, offers a faster adaptation time than WATT-P with a for-loop implementation, albeit at the cost of higher memory usage. It's important to note that methods like DiffTPT [22] and MEMO [14], which show significantly higher adaptation times, employ off-the-shelf diffusion models and AugMix augmentation, respectively, resulting in time-consuming processes that may be impractical for real-world scenarios. In contrast, the effectiveness of our WATT-S method makes it better suited for scenarios where a robust, rapid, and resource-efficient adaptation is crucial.

| Method | Adaptation Time | Memory | Percentage of Learnable Parameters |
|---|---|---|---|
| WATT-S | 2.34 s | 1.5 GB | 0.026% |
| WATT-P | 23.2 s (23.2/8) | 1.5 GB (8 x 1.5 GB) | 0.026% (x8) |
| TENT | 0.28 s | 1.5 GB | 0.026% |
| ClipArTT | 0.55 s | 1.7 GB | 0.026% |
| SAR | 0.42 s | 1.4 GB | 0.026% |
| MEMO | 165 s | 2 GB | 0.026% |
| DiffTPT | 8.2 + 0.26 s | 8.7 GB + 1.7 GB | 0.001% |

Table 7: Comparison of computational cost of different methods.

# E Pseudo-code of our both methods

In Algorithms 1 and 2, we compare the two variants of WATT: one with Parallel MTWA (WATT-P) and the other with Sequential MTWA (WATT-S). The WATT-P model recalibrates its parameters for each template using the average parameters of $m-1$, whereas the WATT-S model updates its parameters solely at the start of each new iteration $m$.

**Algorithm 1** WATT-P - model $f$, parameter $\theta$

**for** $m \in \{1, 2, \ldots, M\}$ **do**
    $\theta_{\text{avg}} \leftarrow \frac{1}{H}\sum_{h=1}^{H}\theta_h$
    **for** $h \in \{1, 2, \ldots, H\}$ **do**
        $f \leftarrow f_{\theta_{\text{avg}}}$
        **for** $l \in \{1, 2, \ldots, L\}$ **do**
            $\theta_h \leftarrow \mathcal{L}_{\text{TTA}}(f_{\theta_{\text{avg}}}(\texttt{template}_h))$
        **end for**
    **end for**
**end for**

**Algorithm 2** WATT-S - model $f$, parameter $\theta$

**for** $m \in \{1, 2, \ldots, M\}$ **do**
    $\theta_{\text{avg}} \leftarrow \frac{1}{H}\sum_{h=1}^{H}\theta_h$
    $f \leftarrow f_{\theta_{\text{avg}}}$
    **for** $h \in \{1, 2, \ldots, H\}$ **do**
        **for** $l \in \{1, 2, \ldots, L\}$ **do**
            $\theta_h \leftarrow \mathcal{L}_{\text{TTA}}(f_{\theta_{\text{avg}}}(\texttt{template}_h))$
        **end for**
    **end for**
**end for**

# F Additional Ablation Studies

**Cross Entropy vs Entropy Minimization.** Two unsupervised loss functions were integrated into previous TTA methods: classical entropy minimization and the loss introduced by CLIPArTT [24], where predictions are utilized as pseudo labels for cross-entropy computation. In Table 8, a comparison between these two loss functions is presented across the original CIFAR-10 dataset and various corruptions from CIFAR-10-C. It is observed that, for these specific corruptions, entropy minimization generally outperforms with the different templates employed, except for *Gaussian Noise*. However, upon assessing the weighted average accuracy, computed after 10 iterations for each template, cross-entropy consistently emerges as the superior option. The marginal impact of the weighted average on entropy minimization suggests that, irrespective of the template used, the model updates in a consistent direction to enhance confidence, rendering cross-entropy the preferred choice for subsequent experiments.

| Dataset | Loss | $T^0$ | $T^1$ | $T^2$ | $T^3$ | $T^4$ | $T^5$ | $T^6$ | $T^7$ | WA |
|---|---|---|---|---|---|---|---|---|---|---|
| CIFAR-10 | TENT | 91.69 | 91.97 | 91.69 | 90.28 | 91.16 | 92.11 | 91.98 | 89.14 | 90.60 |
| | CE | 89.8 | 90.37 | 90.5 | 88.42 | 89.93 | 89.95 | 90.13 | 88.54 | **91.05** |
| Gaussian Noise | TENT | 41.27 | 37.16 | 46.39 | 51.31 | 39.27 | 32.51 | 49.7 | 42.96 | 47.08 |
| | CE | 60.19 | 61.01 | 61.17 | 58.24 | 58.84 | 58.35 | 59.62 | 61.13 | **63.84** |
| Defocus Blur | TENT | 77.12 | 77.13 | 78.7 | 76.09 | 76.85 | 76.59 | 77.86 | 74.31 | 76.21 |
| | CE | 77.23 | 77.07 | 78 | 75.98 | 76.39 | 77.45 | 77.08 | 75.59 | **78.94** |
| Snow | TENT | 78.29 | 79.54 | 80.09 | 75.39 | 78.97 | 78.52 | 78.78 | 75.82 | 77.24 |
| | CE | 76.57 | 77.36 | 77.93 | 75.08 | 77.45 | 77.09 | 77.05 | 75.57 | **79.79** |
| JPEG Compression | TENT | 62.64 | 65.83 | 64.27 | 59.49 | 62.78 | 64.19 | 62.62 | 63.39 | 65.31 |
| | CE | 64.65 | 65.36 | 65.24 | 64.16 | 64.18 | 64.36 | 64.78 | 65.32 | **67.36** |

Table 8: Comparison of accuracy (%) using entropy minimization (TENT) or cross-entropy (CE) on CIFAR-10 and some corruptions of CIFAR-10-C datasets with ViT-B/32 encoder on different templates (please see Fig. 1a) and the weight average.

## G Experiments with another VLM

We investigate if our TTA method is working with other VLMs like SigLip, so we compare if after adaptation with our method the performance improves compared to the baseline in Table 9.

| Dataset | SigLip | WATT-S |
|---|---|---|
| CIFAR-10 | 66.35 | 75.02 ±0.05 |
| CIFAR-10.1 | 57.30 | 65.87 ±0.21 |
| CIFAR-10-C | 37.52 | 45.29 ±0.13 |
| CIFAR-100 | 33.97 | 65.87 ±0.21 |
| CIFAR-100-C | 14.43 | 20.05 ±0.05 |

Table 9: Performance comparison of SigLip and WATT-S on different datasets

## H Experiments on other Visual Encoders

We replicated the experiments from the main paper using alternative visual encoders, ViT-B/16 and ViT-L/14.

| Dataset | Backbone | CLIP | TENT | TPT (BS=32) | CLIPArTT | WATT-P | WATT-S |
|---|---|---|---|---|---|---|---|
| CIFAR-10 | ViT-B/16 | 89.25 | **92.75** ±0.17 | 89.80 ±0.05 | 92.61 ±0.05 | 92.31 ±0.10 | 91.97 ±0.03 |
| | ViT-L/14 | 95.36 | **96.13** ±0.01 | 95.18 ±0.02 | 95.16 ±0.03 | 95.91 ±0.10 | 95.71 ±0.03 |
| CIFAR-10.1 | ViT-B/16 | 84.00 | 88.52 ±0.33 | 83.75 ±0.21 | **88.72** ±0.33 | 87.90 ±0.11 | 88.10 ±0.08 |
| | ViT-L/14 | 91.20 | 92.22 ±0.25 | 91.32 ±0.12 | 91.02 ±0.02 | **92.97** ±0.13 | 92.10 ±0.33 |
| CIFAR-10-C | ViT-B/16 | 60.15 | 68.00 | 59.75 | 73.22 | 75.04 | **76.22** |
| | ViT-L/14 | 76.04 | 79.18 | 75.01 | 78.06 | 80.05 | **80.06** |
| CIFAR-100 | ViT-B/16 | 64.76 | 71.73 ±0.14 | 67.15 ±0.23 | 71.34 ±0.07 | **72.98** ±0.07 | 72.85 ±0.15 |
| | ViT-L/14 | 73.28 | 78.03 ±0.08 | 76.85 ±0.06 | **79.42** ±0.08 | 79.33 ±0.05 | 78.85 ±0.19 |
| CIFAR-100-C | ViT-B/16 | 32.01 | 37.90 | 33.73 | 40.08 | 47.86 | **48.95** |
| | ViT-L/14 | 44.59 | 50.14 | 47.58 | 52.52 | 54.10 | **54.34** |

Table 10: Accuracy (%) on CIFAR-10, CIFAR-10.1, CIFAR-10-C, CIFAR-100 and CIFAR-100-C datasets with ViT-B/16 and ViT-L/14 as visual encoders.

**Performance evaluation in the presence of natural or no domain shift.** As indicated in the main results, employing WATT, both with Parallel and Sequential MTWA, consistently enhances

performance alongside the baseline. This pattern persists across different visual encoders, as shown in Tables 10. Although WATT consistently outperforms the baseline and TPT, TENT and CLIPArTT may occasionally yield superior results depending on the visual encoder used.

**Performance evaluation in the presence of common corruptions.** Table 10 demonstrates a consistent trend where both WATT methods consistently outperform alternative methods across various corruptions and class numbers. Upon closer examination of Table 10, specifically with ViT-B/16 as the visual encoder, Sequential MTWA exhibits a significant performance advantage, surpassing the baseline by 16.07% and the leading method in the *state-of-the-art* by 3.00%. This trend becomes even more pronounced with an increased number of classes, where Sequential MTWA surpasses the baseline and CLIPArTT by 16.94% and 8.87%, respectively.

**Performance analysis under simulated and video shifts.** Our study reveals substantial accuracy improvements on the 3D (simulated shift) and YT (video shift) partitions of VisDA-C when employing different backbones. This enhancement is particularly notable with our proposed WATT method compared to pure CLIP. Notably, the WATT-S variant achieves the highest accuracy across both the 3D and YT partitions, outperforming various adaptation approaches including TENT, TPT, and CLIPArTT. Detailed comparisons can be found in Tables 11 and 12.

**Performance analysis under texture and style shifts.** Findings across the OfficeHome, PACS, and VLCS datasets are detailed in Tables 11 and 12. Across these varied domains, our WATT method demonstrates consistent performance enhancements, as evidenced by both its WATT-P and WATT-S variants. These improvements underscore the efficacy of our approach in mitigating the complexities of texture and style shifts, which pose particular challenges compared to other forms of domain shift.

| Dataset | Domain | CLIP | TENT | TPT | CLIPArTT | WATT-P | WATT-S |
|---|---|---|---|---|---|---|---|
| VisDA-C | 3D (trainset) | 87.16 | 87.57 ±0.01 | 84.04 ±0.03 | 87.58 ±0.00 | 87.61 ±0.01 | **87.72 ±0.02** |
| | YT (valset) | 86.61 | **86.81 ±0.00** | 85.90 ±0.11 | 86.60 ±0.01 | 86.66 ±0.00 | 86.75 ±0.04 |
| | Mean | 86.89 | 87.19 | 84.97 | 87.09 | 87.14 | **87.24** |
| OfficeHome | Art | 79.30 | 79.26 ±0.14 | **81.97 ±0.17** | 79.34 ±0.05 | 80.37 ±0.25 | 80.43 ±0.09 |
| | Clipart | 65.15 | 65.64 ±0.05 | 67.01 ±0.21 | 65.69 ±0.11 | **68.59 ±0.13** | 68.26 ±0.11 |
| | Product | 87.34 | 87.49 ±0.02 | **89.00 ±0.06** | 87.35 ±0.07 | 88.15 ±0.07 | 88.02 ±0.08 |
| | Real World | 89.31 | 89.50 ±0.04 | 89.66 ±0.06 | 89.29 ±0.03 | **90.18 ±0.03** | 90.14 ±0.06 |
| | Mean | 80.28 | 80.47 | **81.91** | 80.42 | 81.82 | 81.71 |
| PACS | Art | 97.44 | 97.54 ±0.02 | 95.10 ±0.41 | 97.64 ±0.02 | 97.49 ±0.08 | **97.66 ±0.08** |
| | Cartoon | 97.38 | 97.37 ±0.04 | 91.42 ±0.22 | 97.37 ±0.02 | 97.47 ±0.04 | **97.51 ±0.02** |
| | Photo | **99.58** | **99.58 ±0.00** | 98.56 ±0.40 | **99.58 ±0.00** | **99.58 ±0.00** | **99.58 ±0.00** |
| | Sketch | 86.06 | 86.37 ±0.05 | 87.23 ±0.06 | 86.79 ±0.04 | **89.73 ±0.16** | 89.56 ±0.14 |
| | Mean | 95.12 | 95.22 | 93.08 | 95.35 | 96.07 | **96.08** |
| VLCS | Caltech101 | **99.43** | **99.43 ±0.00** | 97.62 ±0.12 | **99.43 ±0.00** | 99.36 ±0.00 | 99.36 ±0.00 |
| | LabelMe | 67.75 | 67.31 ±0.14 | 49.77 ±0.03 | 67.74 ±0.10 | 67.55 ±0.39 | **68.59 ±0.25** |
| | SUN09 | 71.74 | 71.57 ±0.15 | 71.56 ±0.86 | 71.67 ±0.01 | 74.75 ±0.07 | **75.16 ±0.12** |
| | VOC2007 | 84.90 | **85.10 ±0.11** | 71.17 ±0.70 | 84.73 ±0.08 | 82.53 ±0.10 | 83.24 ±0.05 |
| | Mean | 80.96 | 80.85 | 72.53 | 80.89 | 81.05 | **81.59** |

Table 11: Accuracy (%) on different domains of VisDA-C, OfficeHome, PACS and VLCS datasets with ViT-B/16 as visual encoder.

| Dataset | Domain | CLIP | TENT | TPT | CLIPArTT | WATT-S |
|---|---|---|---|---|---|---|
| VisDA-C | 3D (trainset) | 91.24 | 91.40 ±0.01 | 90.65 ±0.00 | 91.34 ±0.00 | **91.71 ±0.00** |
| | YT (valset) | 85.62 | 85.77 ±0.01 | 85.41 ±0.06 | 85.61 ±0.01 | **86.80 ±0.01** |
| | Mean | 88.43 | 88.59 | 88.03 | 88.48 | **89.26** |
| OfficeHome | Art | 82.47 | 82.61 ±0.15 | **86.76 ±0.26** | 82.35 ±0.19 | 84.43 ±0.20 |
| | Clipart | 72.20 | 72.51 ±0.03 | 74.76 ±0.07 | 72.41 ±0.06 | **75.43 ±0.08** |
| | Product | 90.94 | 90.97 ±0.02 | **92.42 ±0.07** | 90.94 ±0.06 | 91.88 ±0.05 |
| | Real World | 92.72 | 92.75 ±0.02 | 92.95 ±0.16 | 92.63 ±0.03 | **94.06 ±0.02** |
| | Mean | 84.58 | 84.71 | 86.72 | 84.58 | 86.45 |
| PACS | Art | 98.68 | **98.83 ±0.00** | 94.82 ±0.34 | 98.76 ±0.02 | 98.68 ±0.00 |
| | Cartoon | 97.74 | 97.74 ±0.00 | 95.65 ±0.19 | 97.74 ±0.00 | **97.90 ±0.02** |
| | Photo | 99.54 | 99.54 ±0.03 | 99.44 ±0.03 | **99.54 ±0.00** | **99.64 ±0.00** |
| | Sketch | 93.28 | 93.51 ±0.04 | 92.72 ±0.15 | 93.26 ±0.02 | **93.80 ±0.02** |
| | Mean | 97.31 | 97.41 | 95.66 | 97.33 | **97.51** |
| VLCS | Caltech101 | 99.43 | 99.43 ±0.00 | 97.86 ±0.43 | 99.43 ±0.00 | **99.51 ±0.00** |
| | LabelMe | 69.22 | 69.07 ±0.12 | 52.54 ±0.20 | **69.32 ±0.15** | 62.76 ±0.13 |
| | SUN09 | 68.06 | 68.23 ±0.03 | 69.49 ±0.32 | 67.89 ±0.07 | **72.21 ±0.15** |
| | VOC2007 | 83.99 | 84.08 ±0.15 | 76.16 ±0.63 | **83.89 ±0.13** | 83.02 ±0.12 |
| | Mean | 80.18 | **80.20** | 74.01 | 80.13 | 79.38 |

Table 12: Accuracy (%) on different domains of VisDA-C, OfficeHome, PACS and VLCS datasets with ViT-L/14 as visual encoder.

# I Detailed Experimental Findings

This section provides extensive tables with detailed information on the results, which were summarized in the main body of the paper.

| Dataset | | single_temp | text_avg |
|---|---|---|---|
| CIFAR-10 | | 90.87 ±0.10 | **91.08 ±0.06** |
| CIFAR-10.1 | | 86.80 ±0.19 | **86.85 ±0.18** |
| CIFAR-10-C | Gaussian Noise | 61.20 ±0.05 | **62.09 ±0.15** |
| | Shot noise | 63.16 ±0.09 | **63.51 ±0.03** |
| | Impulse Noise | 55.29 ±0.22 | **56.04 ±0.16** |
| | Defocus blur | 78.03 ±0.12 | **78.66 ±0.07** |
| | Glass blur | 62.7 ±0.24 | **63.35 ±0.25** |
| | Motion blur | 76.33 ±0.11 | **76.96 ±0.14** |
| | Zoom blur | 78.29 ±0.05 | **79.08 ±0.15** |
| | Snow | 78.65 ±0.21 | **78.95 ±0.05** |
| | Frost | 79.49 ±0.11 | **79.95 ±0.06** |
| | Fog | 77.21 ±0.03 | **77.72 ±0.09** |
| | Brightness | 86.60 ±0.07 | **86.98 ±0.06** |
| | Contrast | 79.22 ±0.01 | **79.62 ±0.04** |
| | Elastic transform | 71.17 ±0.25 | **71.61 ±0.09** |
| | Pixelate | 67.59 ±0.08 | **68.70 ±0.15** |
| | JPEG compression | 66.26 ±0.00 | **66.72 ±0.01** |
| | Mean | 72.08 | **72.66** |

Table 13: Accuracy (%) on CIFAR-10, CIFAR-10.1 and CIFAR-10-C datasets with different text ensemble at test time. (WA after 10 iter)×1

| Dataset | | single_temp | text_avg |
|---|---|---|---|
| CIFAR-100 | | 69.79 ±0.20 | **70.30 ±0.11** |
| CIFAR-100-C | Gaussian Noise | 27.17 ±0.22 | **28.08 ±0.21** |
| | Shot noise | 29.69 ±0.20 | **30.47 ±0.19** |
| | Impulse Noise | 25.28 ±0.10 | **26.37 ±0.28** |
| | Defocus blur | 49.83 ±0.11 | **50.52 ±0.04** |
| | Glass blur | 27.83 ±0.03 | **28.25 ±0.06** |
| | Motion blur | 47.77 ±0.21 | **47.89 ±0.21** |
| | Zoom blur | 52.90 ±0.16 | **53.05 ±0.10** |
| | Snow | 50.31 ±0.03 | **50.22 ±0.17** |
| | Frost | 50.79 ±0.07 | **51.08 ±0.09** |
| | Fog | 48.70 ±0.16 | **48.48 ±0.21** |
| | Brightness | 61.22 ±0.06 | **61.56 ±0.16** |
| | Contrast | 47.87 ±0.17 | **47.90 ±0.14** |
| | Elastic transform | 37.55 ±0.13 | **37.93 ±0.19** |
| | Pixelate | 33.81 ±0.06 | **34.56 ±0.14** |
| | JPEG compression | 36.09 ±0.13 | **37.30 ±0.18** |
| | Mean | 41.79 | **42.24** |

Table 14: Accuracy (%) on CIFAR-100 and CIFAR-100-C datasets with different text ensemble at test time. (WA after 10 iter)×1

| Dataset | | Text avg. | Output avg. | Weight avg. (ours) | | |
|---|---|---|---|---|---|---|
| | | | | (after 10 iter)×1 | (after 1 iter)×10 | (after 2 iter)×5 |
| CIFAR-100 | | 69.46 ±0.13 | 70.32 ±0.10 | 70.3 ±0.11 | **70.85 ±0.08** | 70.74 ±0.20 |
| CIFAR-100-C | Gaussian Noise | 27.67 ±0.11 | 28.58 ±0.04 | 28.08 ±0.21 | 31.67 ±0.10 | **32.07 ±0.23** |
| | Shot noise | 30.18 ±0.06 | 31.05 ±0.13 | 30.47 ±0.19 | 34.26 ±0.28 | **34.36 ±0.11** |
| | Impulse Noise | 25.79 ±0.02 | 26.86 ±0.07 | 26.37 ±0.28 | 30.12 ±0.12 | **30.33 ±0.03** |
| | Defocus blur | 49.51 ±0.04 | 51.04 ±0.02 | 50.52 ±0.04 | 52.76 ±0.25 | **52.99 ±0.16** |
| | Glass blur | 27.88 ±0.22 | 28.72 ±0.08 | 28.25 ±0.06 | 31.95 ±0.08 | **32.15 ±0.30** |
| | Motion blur | 46.68 ±0.05 | 48.30 ±0.19 | 47.89 ±0.21 | 50.46 ±0.10 | **50.53 ±0.12** |
| | Zoom blur | 52.05 ±0.07 | 53.72 ±0.11 | 53.05 ±0.10 | 55.13 ±0.29 | **55.30 ±0.22** |
| | Snow | 49.40 ±0.18 | 51.01 ±0.13 | 50.22 ±0.17 | 52.60 ±0.26 | **52.77 ±0.15** |
| | Frost | 49.68 ±0.04 | 51.50 ±0.06 | 51.08 ±0.09 | 53.30 ±0.21 | **53.79 ±0.31** |
| | Fog | 47.36 ±0.17 | 48.67 ±0.22 | 48.48 ±0.21 | 51.35 ±0.08 | **51.49 ±0.21** |
| | Brightness | 60.42 ±0.12 | 61.74 ±0.31 | 61.56 ±0.16 | 63.23 ±0.12 | **63.57 ±0.21** |
| | Contrast | 46.86 ±0.05 | 48.14 ±0.10 | 47.90 ±0.14 | 52.40 ±0.23 | **52.76 ±0.27** |
| | Elastic transform | 37.00 ±0.37 | 38.55 ±0.23 | 37.93 ±0.19 | 40.97 ±0.11 | **40.90 ±0.43** |
| | Pixelate | 33.65 ±0.12 | 34.63 ±0.17 | 34.56 ±0.14 | 40.32 ±0.08 | **40.97 ±0.16** |
| | JPEG compression | 36.38 ±0.11 | 37.67 ±0.23 | 37.30 ±0.18 | 39.35 ±0.19 | **39.59 ±0.08** |
| | Mean | 41.37 | 42.68 | 42.24 | 45.32 | **45.57** |

Table 15: Accuracy (%) on CIFAR-100 and CIFAR-100-C datasets with different averaging

| Dataset | | Text avg. | Output avg. | Weight avg. (ours) | | |
|---|---|---|---|---|---|---|
| | | | | (after 10 iter)×1 | (after 1 iter)×10 | (after 2 iter)×5 |
| CIFAR-10 | | 90.58 ±0.03 | 90.90 ±0.03 | 91.08 ±0.06 | **91.39 ±0.14** | 91.05 ±0.06 |
| CIFAR-10.1 | | 85.78 ±0.25 | 86.77 ±0.08 | 86.85 ±0.18 | **88.02 ±0.18** | 86.98 ±0.31 |
| CIFAR-10-C | Gaussian Noise | 61.23 ±0.13 | 62.22 ±0.12 | 62.09 ±0.15 | 63.42 ±0.07 | **63.84 ±0.24** |
| | Shot noise | 62.88 ±0.15 | 63.98 ±0.17 | 63.51 ±0.03 | 64.93 ±0.13 | **65.28 ±0.21** |
| | Impulse Noise | 54.71 ±0.07 | 56.41 ±0.11 | 56.04 ±0.16 | 58.37 ±0.37 | **58.64 ±0.11** |
| | Defocus blur | 77.93 ±0.12 | 78.63 ±0.18 | 78.66 ±0.07 | **79.11 ±0.17** | 78.94 ±0.12 |
| | Glass blur | 62.37 ±0.18 | 63.32 ±0.07 | 63.35 ±0.25 | 64.67 ±0.18 | **65.12 ±0.07** |
| | Motion blur | 75.55 ±0.19 | 76.97 ±0.05 | 76.96 ±0.14 | 77.56 ±0.12 | **77.81 ±0.14** |
| | Zoom blur | 77.86 ±0.06 | 78.90 ±0.18 | 79.08 ±0.15 | **79.76 ±0.03** | 79.32 ±0.07 |
| | Snow | 77.77 ±0.03 | 78.92 ±0.03 | 78.95 ±0.05 | **79.89 ±0.26** | 79.79 ±0.06 |
| | Frost | 78.51 ±0.09 | 79.67 ±0.09 | 79.95 ±0.06 | 80.52 ±0.04 | **80.54 ±0.12** |
| | Fog | 76.04 ±0.17 | 77.54 ±0.10 | 77.72 ±0.09 | 78.44 ±0.21 | **78.53 ±0.22** |
| | Brightness | 86.08 ±0.13 | 86.75 ±0.04 | 86.98 ±0.06 | **87.32 ±0.14** | 87.11 ±0.11 |
| | Contrast | 77.87 ±0.02 | 79.48 ±0.07 | 79.62 ±0.04 | 80.77 ±0.35 | **81.20 ±0.22** |
| | Elastic transform | 69.98 ±0.16 | 71.20 ±0.22 | 71.61 ±0.09 | 72.52 ±0.19 | **72.66 ±0.15** |
| | Pixelate | 66.78 ±0.29 | 68.27 ±0.17 | 68.70 ±0.15 | 70.50 ±0.20 | **71.11 ±0.13** |
| | JPEG compression | 65.62 ±0.28 | 66.78 ±0.08 | 66.72 ±0.01 | 67.05 ±0.10 | **67.36 ±0.28** |
| | Mean | 71.41 | 72.60 | 72.66 | 73.66 | **73.82** |

Table 16: Accuracy (%) on CIFAR-10, CIFAR-10.1 and CIFAR-10-C datasets with different averaging

| Dataset | | CLIP | BS = 1 | BS = 2 | BS = 4 | BS = 8 | BS = 16 | BS = 32 | BS = 64 | BS = 128 |
|---|---|---|---|---|---|---|---|---|---|---|
| CIFAR-10 | | 88.74 | 89.87 | 89.39 ±0.02 | 89.16 ±0.07 | 88.93 ±0.16 | 89.14 ±0.04 | 89.51 ±0.12 | 90.16 ±0.13 | 91.05 ±0.06 |
| CIFAR-10.1 | | 83.25 | 84.55 | 84.32 ±0.15 | 83.88 ±0.17 | 84.12 ±0.37 | 84.35 ±0.21 | 84.87 ±0.16 | 85.52 ±0.30 | 86.98 ±0.31 |
| CIFAR-10-C | Gaussian Noise | 35.27 | 38.55 | 43.85 ±0.26 | 45.41 ±0.10 | 47.95 ±0.15 | 51.79 ±0.27 | 56.35 ±0.11 | 60.87 ±0.33 | 63.84 ±0.24 |
| | Shot noise | 39.67 | 42.57 | 46.87 ±0.25 | 47.95 ±0.15 | 49.13 ±0.14 | 52.57 ±0.03 | 56.96 ±0.10 | 61.84 ±0.06 | 65.28 ±0.21 |
| | Impulse Noise | 42.61 | 42.92 | 47.94 ±0.29 | 48.20 ±0.18 | 48.69 ±0.11 | 50.53 ±0.18 | 53.32 ±0.19 | 55.81 ±0.11 | 58.64 ±0.11 |
| | Defocus blur | 69.76 | 72.29 | 72.80 ±0.13 | 72.95 ±0.13 | 72.57 ±0.20 | 73.71 ±0.18 | 75.28 ±0.18 | 77.37 ±0.08 | 78.94 ±0.12 |
| | Glass blur | 42.40 | 44.15 | 48.15 ±0.15 | 47.69 ±0.07 | 48.96 ±0.04 | 52.59 ±0.19 | 57.83 ±0.24 | 62.16 ±0.20 | 65.12 ±0.07 |
| | Motion blur | 63.97 | 66.37 | 67.53 ±0.07 | 67.22 ±0.01 | 68.00 ±0.12 | 69.20 ±0.11 | 71.60 ±0.06 | 74.75 ±0.09 | 77.81 ±0.14 |
| | Zoom blur | 69.83 | 71.50 | 72.60 ±0.14 | 72.30 ±0.04 | 72.39 ±0.01 | 73.19 ±0.06 | 75.01 ±0.09 | 77.03 ±0.27 | 79.32 ±0.07 |
| | Snow | 71.78 | 73.72 | 74.46 ±0.16 | 73.97 ±0.19 | 74.12 ±0.05 | 74.62 ±0.22 | 76.06 ±0.06 | 77.64 ±0.06 | 79.79 ±0.06 |
| | Frost | 72.86 | 75.67 | 76.50 ±0.23 | 75.98 ±0.11 | 75.55 ±0.16 | 76.32 ±0.13 | 77.67 ±0.03 | 78.82 ±0.20 | 80.54 ±0.12 |
| | Fog | 67.04 | 68.88 | 70.25 ±0.02 | 69.94 ±0.06 | 69.88 ±0.09 | 71.02 ±0.15 | 73.10 ±0.02 | 75.95 ±0.04 | 78.53 ±0.22 |
| | Brightness | 81.87 | 83.52 | 83.73 ±0.10 | 83.38 ±0.03 | 83.31 ±0.05 | 83.51 ±0.11 | 84.49 ±0.13 | 85.40 ±0.07 | 87.11 ±0.11 |
| | Contrast | 64.37 | 67.02 | 69.67 ±0.13 | 68.64 ±0.14 | 69.08 ±0.06 | 71.11 ±0.17 | 74.58 ±0.14 | 78.25 ±0.22 | 81.20 ±0.22 |
| | Elastic transf. | 60.83 | 62.04 | 64.25 ±0.13 | 63.50 ±0.40 | 63.46 ±0.10 | 64.65 ±0.28 | 66.63 ±0.21 | 69.58 ±0.18 | 72.66 ±0.15 |
| | Pixelate | 50.53 | 51.65 | 55.18 ±0.26 | 55.47 ±0.14 | 56.30 ±0.29 | 58.88 ±0.21 | 63.00 ±0.08 | 67.43 ±0.11 | 71.11 ±0.13 |
| | JPEG compr. | 55.48 | 58.12 | 60.17 ±0.04 | 59.44 ±0.18 | 59.74 ±0.04 | 61.20 ±0.09 | 63.15 ±0.15 | 65.32 ±0.16 | 67.36 ±0.28 |
| | Mean | 59.22 | 61.26 | 63.60 | 63.47 | 63.94 | 65.66 | 68.34 | 71.21 | 73.82 |

Table 17: Accuracy (%) on CIFAR-10, CIFAR-10.1 and CIFAR-10-C datasets with ViT-B/16 as visual encoder for different number of batches.

| Dataset | | T=1 | T=2 | T=4 | T=6 | T=8 |
|---|---|---|---|---|---|---|
| CIFAR-10 | | 89.42 ±0.84 | 90.74 ±0.30 | 90.98 ±0.14 | 91.34 ±0.16 | 91.05 ±0.06 |
| CIFAR-10.1 | | 85.08 ±0.59 | 86.49 ±0.59 | 87.20 ±0.48 | 87.53 ±0.21 | 86.98 ±0.31 |
| CIFAR-10-C | Gaussian Noise | 59.82 ±1.43 | 62.05 ±0.62 | 62.79 ±0.18 | 63.49 ±0.27 | 63.84 ±0.24 |
| | Shot noise | 62.32 ±1.32 | 63.35 ±0.43 | 64.73 ±0.31 | 65.02 ±0.10 | 65.28 ±0.21 |
| | Impulse Noise | 54.07 ±0.17 | 56.83 ±0.33 | 57.53 ±0.47 | 58.37 ±0.08 | 58.64 ±0.11 |
| | Defocus blur | 77.09 ±0.24 | 78.32 ±0.32 | 78.92 ±0.16 | 79.17 ±0.26 | 78.94 ±0.12 |
| | Glass blur | 60.64 ±0.29 | 63.77 ±0.43 | 64.42 ±0.68 | 64.64 ±0.39 | 65.12 ±0.07 |
| | Motion blur | 74.60 ±0.50 | 77.02 ±0.32 | 77.70 ±0.26 | 77.73 ±0.12 | 77.81 ±0.14 |
| | Zoom blur | 77.40 ±0.29 | 78.93 ±0.46 | 79.28 ±0.54 | 79.33 ±0.24 | 79.32 ±0.07 |
| | Snow | 76.96 ±1.04 | 78.83 ±0.31 | 79.47 ±0.29 | 79.69 ±0.33 | 79.79 ±0.06 |
| | Frost | 77.62 ±0.86 | 79.27 ±0.45 | 80.04 ±0.26 | 80.46 ±0.17 | 80.54 ±0.12 |
| | Fog | 75.32 ±0.57 | 77.27 ±0.39 | 78.00 ±0.17 | 78.55 ±0.29 | 78.53 ±0.22 |
| | Brightness | 85.13 ±0.58 | 86.74 ±0.22 | 87.07 ±0.20 | 87.13 ±0.21 | 87.11 ±0.11 |
| | Contrast | 77.18 ±0.68 | 79.74 ±0.31 | 80.32 ±0.07 | 80.69 ±0.12 | 81.20 ±0.22 |
| | Elastic transform | 69.39 ±0.39 | 71.40 ±0.24 | 72.25 ±0.14 | 72.28 ±0.34 | 72.66 ±0.15 |
| | Pixelate | 66.26 ±0.76 | 68.86 ±0.68 | 69.47 ±0.39 | 71.00 ±0.57 | 71.11 ±0.13 |
| | JPEG compression | 64.58 ±0.58 | 66.28 ±0.14 | 66.82 ±0.24 | 67.16 ±0.18 | 67.36 ±0.28 |
| | Mean | 70.56 | 72.58 | 73.25 | 73.65 | 73.82 |

Table 18: Accuracy (%) on CIFAR-10, CIFAR-10.1 and CIFAR-10-C datasets with ViT-B/16 as visual encoder for different number of templates randomly picked over 5 runs.

| Dataset | | CLIP | TENT | TPT (BS=32) | CLIPArTT | WATT-P | WATT-S |
|---|---|---|---|---|---|---|---|
| CIFAR-10 | | 88.74 | **91.69 ±0.10** | 88.06 ±0.06 | 90.04 ±0.13 | 91.41 ±0.17 | 91.05 ±0.06 |
| CIFAR-10.1 | | 83.25 | 87.60 ±0.45 | 81.80 ±0.27 | 86.35 ±0.27 | **87.78 ±0.05** | 86.98 ±0.31 |
| CIFAR-10-C | Gaussian Noise | 35.27 | 41.27 ±0.27 | 33.90 ±0.08 | 59.90 ±0.36 | 61.89 ±0.24 | **63.84 ±0.24** |
| | Shot noise | 39.67 | 47.20 ±0.23 | 38.20 ±0.02 | 62.77 ±0.07 | 63.52 ±0.08 | **65.28 ±0.21** |
| | Impulse Noise | 42.61 | 48.58 ±0.31 | 37.66 ±0.20 | 56.02 ±0.16 | 57.13 ±0.02 | **58.64 ±0.11** |
| | Defocus blur | 69.76 | 77.12 ±0.16 | 67.83 ±0.28 | 76.74 ±0.05 | 78.86 ±0.09 | **78.94 ±0.12** |
| | Glass blur | 42.40 | 52.65 ±0.30 | 38.81 ±0.12 | 61.77 ±0.16 | 62.88 ±0.06 | **65.12 ±0.07** |
| | Motion blur | 63.97 | 71.25 ±0.09 | 63.39 ±0.13 | 76.01 ±0.19 | 76.85 ±0.26 | **77.81 ±0.14** |
| | Zoom blur | 69.83 | 76.20 ±0.19 | 68.95 ±0.16 | 77.40 ±0.20 | **79.35 ±0.04** | 79.32 ±0.07 |
| | Snow | 71.78 | 78.29 ±0.20 | 70.16 ±0.10 | 77.29 ±0.16 | 79.44 ±0.09 | **79.79 ±0.06** |
| | Frost | 72.86 | 79.84 ±0.09 | 72.39 ±0.22 | 79.20 ±0.08 | 80.13 ±0.10 | **80.54 ±0.12** |
| | Fog | 67.04 | 77.39 ±0.01 | 64.31 ±0.28 | 75.74 ±0.14 | 77.68 ±0.07 | **78.53 ±0.22** |
| | Brightness | 81.87 | **87.78 ±0.03** | 81.30 ±0.18 | 86.59 ±0.16 | 87.10 ±0.10 | 87.11 ±0.11 |
| | Contrast | 64.37 | 79.47 ±0.11 | 62.26 ±0.31 | 77.82 ±0.14 | 80.04 ±0.24 | **81.20 ±0.22** |
| | Elastic transform | 60.83 | 70.00 ±0.25 | 56.43 ±0.27 | 70.20 ±0.01 | 71.76 ±0.10 | **72.66 ±0.15** |
| | Pixelate | 50.53 | 63.74 ±0.18 | 42.80 ±0.40 | 66.52 ±0.13 | 69.28 ±0.09 | **71.11 ±0.13** |
| | JPEG compression | 55.48 | 62.64 ±0.14 | 53.67 ±0.25 | 63.51 ±0.14 | 66.49 ±0.14 | **67.36 ±0.28** |
| | Mean | 59.22 | 67.56 | 56.80 | 71.17 | 72.83 | **73.82** |

Table 19: Accuracy (%) on CIFAR-10, CIFAR-10.1 and CIFAR-10-C datasets with ViT-B/32 as visual encoder.

| Dataset | | CLIP | TENT | TPT (BS=32) | CLIPArTT | WATT-P | WATT-S |
|---|---|---|---|---|---|---|---|
| CIFAR-10 | | 89.25 | 92.75 ±0.17 | 89.80 ±0.05 | 92.61 ±0.05 | 92.31 ±0.10 | 91.97 ±0.03 |
| CIFAR-10.1 | | 84.00 | 88.52 ±0.33 | 83.75 ±0.21 | 88.72 ±0.33 | 87.9 ±0.11 | 88.10 ±0.08 |
| CIFAR-10-C | Gaussian Noise | 37.75 | 31.04 ±0.38 | 35.35 ±0.15 | 60.89 ±0.26 | 63.10 ±0.12 | 65.57 ±0.22 |
| | Shot noise | 41.10 | 40.54 ±0.41 | 41.03 ±0.19 | 65.19 ±0.21 | 66.31 ±0.10 | 68.67 ±0.37 |
| | Impulse Noise | 51.71 | 58.03 ±0.16 | 54.86 ±0.07 | 67.55 ±0.09 | 69.62 ±0.12 | 70.39 ±0.11 |
| | Defocus blur | 70.07 | 77.57 ±0.03 | 70.29 ±0.02 | 78.92 ±0.12 | 79.64 ±0.08 | 79.90 ±0.07 |
| | Glass blur | 42.24 | 47.16 ±0.05 | 37.86 ±0.17 | 57.18 ±0.20 | 58.98 ±0.12 | 61.62 ±0.21 |
| | Motion blur | 65.81 | 76.16 ±0.05 | 67.43 ±0.11 | 76.59 ±0.06 | 78.32 ±0.16 | 79.02 ±0.07 |
| | Zoom blur | 72.50 | 79.64 ±0.12 | 72.91 ±0.02 | 79.62 ±0.11 | 80.67 ±0.07 | 81.10 ±0.08 |
| | Snow | 73.23 | 81.68 ±0.03 | 72.98 ±0.32 | 81.13 ±0.29 | 81.99 ±0.10 | 82.54 ±0.18 |
| | Frost | 76.52 | 83.22 ±0.05 | 75.87 ±0.16 | 81.24 ±0.08 | 83.41 ±0.16 | 83.46 ±0.15 |
| | Fog | 68.35 | 80.78 ±0.15 | 69.13 ±0.27 | 78.47 ±0.19 | 81.36 ±0.12 | 81.88 ±0.12 |
| | Brightness | 83.36 | 89.85 ±0.11 | 83.67 ±0.14 | 88.66 ±0.15 | 89.06 ±0.05 | 89.10 ±0.14 |
| | Contrast | 61.90 | 79.24 ±0.19 | 62.16 ±0.06 | 75.15 ±0.07 | 81.57 ±0.23 | 83.79 ±0.12 |
| | Elastic transform | 53.16 | 62.54 ±0.08 | 51.26 ±0.23 | 69.49 ±0.08 | 69.14 ±0.09 | 70.93 ±0.20 |
| | Pixelate | 48.48 | 67.08 ±0.24 | 44.65 ±0.21 | 71.80 ±0.16 | 73.38 ±0.29 | 75.67 ±0.32 |
| | JPEG compression | 56.05 | 65.42 ±0.05 | 56.73 ±0.07 | 66.42 ±0.25 | 69.02 ±0.10 | 69.65 ±0.23 |
| | Mean | 60.15 | 68.00 | 59.75 | 73.22 | 75.04 | 76.22 |

Table 20: Accuracy (%) on CIFAR-10, CIFAR-10.1 and CIFAR-10-C datasets with ViT-B/16 as visual encoder.

| Dataset | | CLIP | TENT | TPT (BS=32) | CLIPArTT | WATT-P | WATT-S |
|---|---|---|---|---|---|---|---|
| CIFAR-100 | | 64.76 | 71.73 ±0.14 | 67.15 ±0.23 | 71.34 ±0.07 | 72.98 ±0.07 | 72.85 ±0.15 |
| CIFAR-100-C | Gaussian Noise | 15.88 | 12.28 ±0.20 | 15.43 ±0.03 | 19.01 ±0.24 | 34.23 ±0.03 | 35.95 ±0.27 |
| | Shot noise | 17.49 | 15.07 ±0.21 | 16.88 ±0.07 | 20.27 ±0.21 | 36.68 ±0.1 | 37.96 ±0.15 |
| | Impulse Noise | 21.43 | 13.13 ±0.16 | 22.12 ±0.15 | 17.66 ±0.10 | 43.17 ±0.35 | 44.62 ±0.2 |
| | Defocus blur | 40.10 | 50.35 ±0.03 | 41.08 ±0.22 | 49.86 ±0.13 | 53.13 ±0.12 | 53.80 ±0.12 |
| | Glass blur | 13.48 | 4.84 ±0.14 | 18.43 ±0.15 | 18.34 ±0.31 | 32.53 ±0.03 | 33.39 ±0.11 |
| | Motion blur | 39.82 | 49.85 ±0.37 | 40.85 ±0.26 | 50.00 ±0.09 | 51.63 ±0.06 | 52.72 ±0.30 |
| | Zoom blur | 45.45 | 54.76 ±0.04 | 46.77 ±0.06 | 54.13 ±0.08 | 56.81 ±0.11 | 57.51 ±0.09 |
| | Snow | 42.77 | 52.38 ±0.18 | 47.24 ±0.18 | 52.80 ±0.27 | 56.04 ±0.06 | 56.73 ±0.27 |
| | Frost | 45.39 | 51.66 ±0.04 | 48.61 ±0.14 | 49.56 ±0.08 | 56.00 ±0.11 | 56.48 ±0.34 |
| | Fog | 38.98 | 50.74 ±0.14 | 39.92 ±0.16 | 49.92 ±0.11 | 52.88 ±0.33 | 53.83 ±0.19 |
| | Brightness | 52.55 | 64.26 ±0.09 | 55.83 ±0.10 | 63.76 ±0.13 | 65.58 ±0.07 | 66.67 ±0.19 |
| | Contrast | 33.32 | 48.69 ±0.08 | 33.13 ±0.16 | 47.86 ±0.02 | 52.90 ±0.06 | 55.06 ±0.15 |
| | Elastic transform | 24.39 | 33.56 ±0.28 | 27.36 ±0.10 | 32.93 ±0.23 | 39.82 ±0.21 | 40.37 ±0.26 |
| | Pixelate | 21.89 | 36.20 ±0.28 | 21.26 ±0.10 | 39.49 ±0.21 | 45.10 ±0.06 | 47.02 ±0.04 |
| | JPEG compression | 27.21 | 30.80 ±0.05 | 30.97 ±0.10 | 35.56 ±0.23 | 41.43 ±0.18 | 42.13 ±0.21 |
| | Mean | 32.01 | 37.90 | 33.73 | 40.08 | 47.86 | 48.95 |

Table 21: Accuracy (%) on CIFAR-100 and CIFAR-100-C datasets with ViT-B/16 as visual encoder.

| Dataset | | CLIP | TENT | TPT (BS=32) | CLIPArTT | WATT-P | WATT-S |
|---|---|---|---|---|---|---|---|
| CIFAR-10 | | 95.36 | 96.13 ±0.06 | 95.18 ±0.02 | 95.16 ±0.03 | 95.91 ±0.10 | 95.71 ±0.03 |
| CIFAR-10.1 | | 91.20 | 92.22 ±0.25 | 91.32 ±0.12 | 91.02 ±0.02 | 92.97 ±0.13 | 92.10 ±0.33 |
| CIFAR-10-C | Gaussian Noise | 64.64 | 68.87 ±0.20 | 64.44 ±0.11 | 70.04 ±0.31 | 72.81 ±0.09 | 72.73 ±0.03 |
| | Shot noise | 67.82 | 71.95 ±0.06 | 66.81 ±0.19 | 71.44 ±0.16 | 74.45 ±0.16 | 74.60 ±0.03 |
| | Impulse Noise | 78.21 | 80.22 ±0.19 | 76.46 ±0.17 | 79.42 ±0.15 | 81.36 ±0.09 | 80.95 ±0.15 |
| | Defocus blur | 80.73 | 83.10 ±0.03 | 79.01 ±0.23 | 81.75 ±0.19 | 83.20 ±0.10 | 83.15 ±0.18 |
| | Glass blur | 50.29 | 57.12 ±0.07 | 49.64 ±0.23 | 58.13 ±0.23 | 61.51 ±0.07 | 62.35 ±0.15 |
| | Motion blur | 80.75 | 82.69 ±0.11 | 78.85 ±0.04 | 80.76 ±0.12 | 82.60 ±0.13 | 82.61 ±0.12 |
| | Zoom blur | 82.75 | 84.91 ±0.08 | 82.32 ±0.13 | 83.39 ±0.05 | 85.76 ±0.06 | 85.44 ±0.13 |
| | Snow | 83.01 | 85.99 ±0.11 | 82.69 ±0.10 | 84.48 ±0.07 | 84.91 ±0.13 | 85.61 ±0.15 |
| | Frost | 84.90 | 87.15 ±0.12 | 84.63 ±0.08 | 85.21 ±0.06 | 87.17 ±0.13 | 86.88 ±0.04 |
| | Fog | 78.44 | 81.30 ±0.07 | 77.56 ±0.17 | 79.27 ±0.07 | 81.80 ±0.11 | 81.79 ±0.09 |
| | Brightness | 91.67 | 93.07 ±0.04 | 90.94 ±0.04 | 91.87 ±0.09 | 92.78 ±0.05 | 92.59 ±0.16 |
| | Contrast | 84.20 | 87.93 ±0.04 | 82.88 ±0.09 | 86.19 ±0.06 | 87.54 ±0.12 | 87.38 ±0.02 |
| | Elastic transform | 65.45 | 69.96 ±0.12 | 64.81 ±0.14 | 67.43 ±0.24 | 71.19 ±0.07 | 71.25 ±0.09 |
| | Pixelate | 75.10 | 77.89 ±0.05 | 72.92 ±0.12 | 77.11 ±0.10 | 77.88 ±0.13 | 77.67 ±0.16 |
| | JPEG compression | 72.58 | 75.49 ±0.07 | 71.18 ±0.19 | 74.46 ±0.11 | 75.88 ±0.16 | 75.84 ±0.18 |
| | Mean | 76.04 | 79.18 | 75.01 | 78.06 | 80.05 | 80.06 |

Table 22: Accuracy (%) on CIFAR-10, CIFAR-10.1 and CIFAR-10-C datasets with ViT-L/14 as visual encoder.

| Dataset | | CLIP | TENT | TPT,(BS=32) | CLIPArTT | WATT-P | WATT-S |
|---|---|---|---|---|---|---|---|
| CIFAR-100 | | 73.28 | 78.03 ±0.08 | 76.85 ±0.06 | 79.42 ±0.08 | 79.33 ±0.05 | 78.85 ±0.19 |
| CIFAR-100-C | Gaussian Noise | 30.55 | 36.93 ±0.03 | 36.10 ±0.11 | 41.46 ±0.15 | 43.99 ±0.13 | 44.13 ±0.11 |
| | Shot noise | 34.58 | 40.96 ±0.16 | 38.23 ±0.13 | 44.27 ±0.09 | 46.28 ±0.22 | 46.63 ±0.17 |
| | Impulse Noise | 44.89 | 49.09 ±0.14 | 49.69 ±0.21 | 51.44 ±0.23 | 56.15 ±0.04 | 56.26 ±0.22 |
| | Defocus blur | 48.88 | 55.23 ±0.07 | 50.43 ±0.19 | 56.55 ±0.22 | 57.46 ±0.01 | 57.66 ±0.42 |
| | Glass blur | 23.46 | 27.02 ±0.23 | 24.35 ±0.22 | 30.47 ±0.14 | 32.54 ±0.12 | 33.54 ±0.16 |
| | Motion blur | 50.83 | 56.03 ±0.20 | 51.94 ±0.04 | 56.98 ±0.18 | 58.22 ±0.10 | 57.81 ±0.05 |
| | Zoom blur | 56.02 | 61.19 ±0.10 | 56.96 ±0.16 | 62.56 ±0.04 | 62.94 ±0.02 | 62.74 ±0.06 |
| | Snow | 49.03 | 55.60 ±0.09 | 54.89 ±0.11 | 58.81 ±0.11 | 60.68 ±0.06 | 61.04 ±0.27 |
| | Frost | 53.27 | 58.21 ±0.15 | 58.15 ±0.33 | 60.38 ±0.23 | 62.34 ±0.14 | 62.76 ±0.22 |
| | Fog | 48.51 | 53.37 ±0.25 | 49.26 ±0.13 | 54.38 ±0.04 | 54.71 ±0.31 | 54.70 ±0.13 |
| | Brightness | 60.53 | 67.34 ±0.17 | 66.60 ±0.10 | 69.63 ±0.14 | 71.52 ±0.07 | 71.60 ±0.09 |
| | Contrast | 50.24 | 59.91 ±0.13 | 53.64 ±0.24 | 63.39 ±0.13 | 62.77 ±0.22 | 63.95 ±0.04 |
| | Elastic transform | 35.07 | 38.49 ±0.12 | 35.72 ±0.09 | 39.57 ±0.39 | 41.28 ±0.25 | 41.27 ±0.15 |
| | Pixelate | 43.86 | 48.37 ±0.17 | 44.32 ±0.10 | 50.45 ±0.16 | 51.15 ±0.15 | 51.22 ±0.13 |
| | JPEG compression | 39.11 | 44.42 ±0.09 | 43.44 ±0.11 | 47.45 ±0.14 | 49.40 ±0.17 | 49.78 ±0.18 |
| | Mean | 44.59 | 50.14 | 47.58 | 52.52 | 54.10 | 54.34 |

Table 23: Accuracy (%) on CIFAR-100 and CIFAR-100-C datasets with ViT-L/14 as visual encoder.

