# OpenReview forum: "WATT: Weight Average Test Time Adaptation of CLIP"
_NeurIPS.cc/2024/Conference — NeurIPS 2024 poster_

### Official Review · Reviewer_HDdi · 2024-07-09

**Soundness:** 3
**Presentation:** 3
**Contribution:** 2
**Rating:** 8
**Confidence:** 4

**Summary:**

The authors introduce a Test-Time Adaptation technique named Weight Average Test-Time Adaptation (WATT) for Vision-Language Models (VLMs) such as CLIP. WATT is the proposed method that improves test-time adaptation by borrowing ideas from different text prompt templates to develop pseudo-labels for model updates and also carrying out weight averaging so as to consolidate learned information from those generated labels. The approach is tested across different datasets, revealing its capacity to boost performance without any further metamorphosis of the model or additional trainable modules.

**Strengths:**

+ The novelty of the study is that it proposes a novel test-time approach. It involves averaging weights while using varying text prompts which deviates from the conventional TTA techniques— making it a high-quality extension to existing TTA methods.
+ It ensures quality by using rigorous experimental setup: involving thorough evaluations on various datasets, with good performance demonstrated by improvements over the current leading techniques.
+ It maintains clarity in its presentation: both of methodology and results. This has been made possible by visual aids and detailed elaborations that ensure better understanding, thus though we still need to work more on this document to achieve perfection.
+ The importance of the proposed method lies in its ability to adapt using a single image without requiring any further changes to the model; this level of adaptability is highly significant for real-world applications across different fields.

**Weaknesses:**

- Limited Scope of Evaluation: State-of-the-art Methods [1, 2] published in recent CVPR/ICCV are missing for comparing in Experimental part and discussing in the Related Work section.
[1] Diverse data augmentation with diffusions for effective test-time prompt tuning. ICCV, 2023.
[2] Efficient Test-Time Adaptation of Vision-Language Models. CVPR, 2024.
- Computational Cost: The paper mentions the efficiency of WATT but does not provide a detailed comparison of computational costs with other TTA methods, which could be crucial for practical deployment.
- Generalization to Other Tasks: The paper focuses on image classification tasks. Extending the approach to other vision tasks like segmentation or detection and evaluating its performance there could provide a more comprehensive understanding of its applicability.

**Questions:**

1. Could you provide a more detailed comparison of the computational cost of WATT compared to other TTA methods?
2. Have you considered evaluating WATT on real-world datasets with more complex domain shifts?
3. How would WATT perform on other vision tasks such as segmentation or object detection?

**Limitations:**

The authors have addressed the limitations and potential negative societal impacts adequately. They highlight the focus on image classification and suggest future work extending the method to other tasks. However, they could discuss more on the computational resource requirements and scalability in practical deployments.

---

> ### Author Rebuttal · Authors · 2024-08-07
>
> ***Q:* Limited Scope of Evaluation.**
>
> *A:* Thank you for your valuable feedback regarding the scope of our evaluation. We appreciate the recommendation to include comparisons with more recent state-of-the-art methods in both the Experimental section and the Related Work section. Specifically, we have taken note of the methods mentioned: DiffTPT [Diverse Data Augmentation with Diffusions for Effective Test-Time Prompt Tuning, ICCV 2023] and TDA [Efficient Test-Time Adaptation of Vision-Language Models, CVPR 2024]. Therefore, we compared them with the TTA method mentioned in our paper and SAR as noted by Reviewer y4Me. The results are summarized in the table below:
>
> | Dataset       | **CLIP** | **TENT** | **TPT** | **CLIPArTT** | **TDA** | **DiffTPT** | **SAR** | **WATT-P** | **WATT-S** |
> | :------------ | :------- | :------- | :------ | :----------- | :------ | :---------- | :------ | :--------- | :--------- |
> | **CIFAR-10**  | 88.74    | 91.69 &plusmn; 0.10 | 88.06 &plusmn; 0.06 | 90.04 &plusmn; 0.13 | 84.09 &plusmn; 0.04 | 83.07 &plusmn; 0.05 | 89.05 &plusmn; 0.02 | 91.41 &plusmn; 0.17 | 91.05 &plusmn; 0.06 |
> | **CIFAR-10.1**| 83.25    | 87.60 &plusmn; 0.45 | 81.80 &plusmn; 0.27 | 86.35 &plusmn; 0.27 | 78.98 &plusmn; 0.37 | 76.50 &plusmn; 0.29 | 83.65 &plusmn; 0.04 | 87.78 &plusmn; 0.05 | 86.98 &plusmn; 0.31 |
> | **CIFAR-10-C**| 59.22    | 67.56              | 56.80            | 71.17              | 48.00            | 56.77            | 60.45            | 72.83            | 73.82            |
> | **CIFAR-100** | 61.68    | 69.74 &plusmn; 0.16 | 63.78 &plusmn; 0.28 | 69.79 &plusmn; 0.04 | 60.32 &plusmn; 0.06 | 52.80 &plusmn; 0.08 | 64.44 &plusmn; 0.01 | 70.38 &plusmn; 0.14 | 70.74 &plusmn; 0.20 |
> | **CIFAR-100-C**| 29.43    | 35.19              | 30.46            | 41.51              | 22.08            | 22.89            | 31.92            | 44.68            | 45.57            |
>
> According to the TDA supplementary materials, we selected the weighting factor alpha as 5.0 and the sharpness ratio beta as 2.0, which are stated as optimal. However, these values did not appear to be the best choice for more challenging datasets like CIFAR-10-C or CIFAR-100-C, which contain various corruptions. Adjusting these parameters based on the dataset would not be consistent with the principles of a fully TTA method, which might explain their suboptimal performance in our results.
> Regarding DiffTPT, it generates 64 images per test image, making it challenging to use in a real-world TTA scenario. Similar to TDA, DiffTPT requires carefully chosen parameters to fit the dataset, whereas our method does not require dataset-specific tuning. This highlights the robustness and practicality of our approach in diverse real-world applications.
>
> ***Q:* Computational Cost.**
>
> *A:* To address the reviewer's concern regarding the computational cost comparison of WATT with other TTA methods, we have conducted a thorough evaluation under consistent conditions using an NVIDIA A6000 GPU within the same Python environment. The table provided compares the adaptation time, memory usage, and the number of learnable parameters for various TTA methods, including our proposed WATT method.The table clearly demonstrates that WATT-S, a sequential implementation of WATT, maintains competitive adaptation time and memory usage compared to other methods like TENT and ClipArTT, which are efficient but lack the robustness of WATT's method. Additionally, the table highlights that WATT-P, with parallel model training, offers a faster adaptation time than WATT-P with a for-loop implementation, albeit at the cost of higher memory usage.
> It's important to note that methods like DiffTPT and MEMO, which show significantly higher adaptation times, employ off-the-shelf diffusion models and AugMix augmentation, respectively, resulting in time-consuming processes that may be impractical for real-world scenarios. In contrast, the effectiveness of our WATT-S method makes it better suited for scenarios where a robust, rapid, and resource-efficient adaptation is crucial.
>
> | Method     | **Adaptation Time** | **GPU Memory**      | **Percentage of  Learnable Parameters** |
> | :--------- | :------------------ | :-------------- | :------------------------- |
> | **TENT**   | 0.3 sec.             | 1.5 GB          | 0.026%                     |
> | **ClipArTT**| 0.6 sec.             | 1.7 GB          | 0.026%                     |
> | **SAR**    | 0.4 sec.              | 1.4 GB          | 0.026%                     |
> | **MEMO**   | 165 sec.               | 2 GB            | 0.026%                     |
> | **DiffTPT**| 8.5 sec.          | 10 GB |         0.001%                    |
> | **WATT-P** | 23.2 sec.     | 1.5 GB  | 0.026%            |
> | **WATT-P (concurrent)** | 2.9 sec.     | 10 GB | 0.208%              |
> | **WATT-S** | 2.3 sec.              | 1.5 GB          | 0.026%                     |
>
> We noticed that we used the word "efficiency" in lines 105 and 382. We apologize for this and should have used "effectively" instead.
>
> ***Q:* Generalization to Other Tasks.**
>
> *A:* While our current work focuses on image classification, exploring the applicability of WATT to tasks such as segmentation and object detection is indeed an interesting direction for future research. It is worth noting that many recent TTA papers also primarily focus on classification tasks. Extending to other tasks would require tailored experimental settings and potentially different methodological adjustments (e.g., CLIP cannot be used directly for segmentation), which we believe are beyond the scope of this paper. We appreciate your suggestion and consider it a valuable avenue for further investigation.
>
> ***Q:* Have you considered evaluating WATT on real-world datasets with more complex domain shifts?**
>
> *A:* Due to limited space, please refer to the global rebuttal for a general explanation of the datasets we used.

---

> ### Comment · Reviewer_HDdi · 2024-08-08
>
> Thank you for thoroughly addressing my concerns. After reviewing the comments and responses for the other reviewers, I see that their concerns have also been resolved. The authors have provided clear definitions of terms for better understanding, conducted additional experiments to further evaluate the effectiveness of the proposed method, and offered more in-depth analysis of how the proposed method works in various settings.
>
> Overall, the rebuttal enhances my confidence in this paper. With careful consideration, I believe this paper with revision is worthy of NeurIPS and will significantly impact the test-time adaptation field. My final decision is “strong accept.”

---

> > ### Author Response · Authors · 2024-08-08
> >
> > Thank you for your thorough review and constructive feedback throughout this process. We greatly appreciate your positive evaluation and are pleased that our revisions have addressed your concerns.

---

### Official Review · Reviewer_y4Me · 2024-07-10

**Soundness:** 3
**Presentation:** 3
**Contribution:** 3
**Rating:** 6
**Confidence:** 4

**Summary:**

This paper proposes a test-time adaptation (TTA) method for CLIP by integrating various textual prompt templates into Weight Average (WA) methods (Ref [11], [22]). Experiments on multiple types of domain shifts show the effectiveness of the proposed method.

**Strengths:**

- The proposed method is effective yet simple.
- The experiments span different domain shift scenarios and demonstrate better performance of the method.

**Weaknesses:**

- The second contribution at the end of Section 1 lacks sufficient evidence. Table 4 only shows that the proposed method can handle small batch sizes without comparison with others. Additionally, there are existing TTA works studying small batch size scenarios. (SAR [Towards Stable Test-Time Adaptation in Dynamic Wild World, 2023], MEMO [MEMO: Test Time Robustness via Adaptation and Augmentation, 2022], etc.). Recent diffusion model-based TTA methods (such as DDA [Back to the Source: Diffusion-Driven Test-Time Adaptation, 2023]) also show robustness for small batch sizes.
- The technical contribution seems not that significant. The core idea is introducing multiple prompt templates into two previous WA methods, to my understanding.

**Questions:**

Is the evaluation done online (like the setting in the TENT paper where inference and adaptation happen on each batch in data streams), or does the model adapt with a single batch and then perform inference on the whole test set?

**Limitations:**

The authors mention limitations in the checklist.

---

> ### Author Rebuttal · Authors · 2024-08-07
>
> ***Q:* Comparison with other methods and a batch size of 1.**
>
> *A:* Thank you for your insightful feedback. We appreciate your comments regarding the need for more comprehensive evidence to support our second contribution in Section 1. In response, we have tested the TTA methods mentioned in our paper with a batch size of 1. Additionally, we implemented the comparisons you suggested and adapted SAR [Towards Stable Test-Time Adaptation in Dynamic Wild World, 2023] and MEMO [MEMO: Test Time Robustness via Adaptation and Augmentation, 2022] to CLIP.
> We decided not to implement DDA [Back to the Source: Diffusion-Driven Test-Time Adaptation, 2023]. While it is an excellent article, its benchmarks differ from ours. In DDA, the input is adapted rather than the encoder, and it aligns more with test-time training than fully test-time adaptation, as it requires the source dataset to train the diffusion model.
>
> The results with a batch size of 1 are summarized in the table below:
>
> | Dataset       | **CLIP** | **TPT** | **CLIPARTT** | **SAR** | **MEMO** | **WATT-S** |
> | :------------ | :------- | :------ | :----------- | :------ | :------- | :--------- |
> | **CIFAR-10**  | 88.74    | 88.29   | 88.76        | 87.41   | 89.29    | 89.87      |
> | **CIFAR 10.1**| 83.25    | 82.85   | 83.15        | 82.32   | 83.80    | 84.55      |
> | **CIFAR-10-C**| 59.22    | 59.03   | 59.18        | 58.70   | 61.15    | 61.26      |
>
> As can be seen, WATT-S achieves the highest accuracy in all cases. On CIFAR-10.1, it outperforms SAR by 2.23% and MEMO by 0.75%. As highlighted in our paper, this improvement is achieved without any image augmentation, a common practice in previous TTA approaches working with small batches.
>
> ***Q:* The technical contribution seems not that significant. The core idea is introducing multiple prompt templates into two previous WA methods, to my understanding.**
>
> *A:* Our work introduces the use of multiple prompt templates in weight averaging (WA) methods, which, to our knowledge, is a novel approach in the TTA field. As noted by Reviewer HDdi, this approach deviates from conventional TTA techniques and offers a valuable extension to existing methods. We have also conducted a comparative analysis of WA with other common ensembling methods to highlight its effectiveness.
> Moreover, our paper is the first to demonstrate performance fluctuations due to using different text templates. Recognizing this, we leveraged these fluctuations as a benefit to enhance WA, whereas previous articles focused on using image augmentations or varying hyperparameters in the training time.
>
> ***Q:* Is the evaluation done online (like the setting in the TENT paper where inference and adaptation happen on each batch in data streams), or does the model adapt with a single batch and then perform inference on the whole test set?**
>
> *A:* In the context of the TENT paper, the model adapts to a given batch, and updates its parameters from the most recent adaptation when a new batch is introduced. In this framework, adaptation and inference occur within the same batch, similar to TENT. However, upon introducing a new batch, the model reverts to the standard CLIP configuration before making the adaptation.

---

> > ### Comment · Reviewer_y4Me · 2024-08-12
> >
> > Thank you for the additional experiments and explanations. The experiments have addressed my main concern, so I have raised my score from 5 to 6. However, I still maintain that the technical contribution is not particularly significant.

---

### Official Review · Reviewer_wUmo · 2024-07-12

**Soundness:** 4
**Presentation:** 3
**Contribution:** 3
**Rating:** 7
**Confidence:** 3

**Summary:**

This paper proposes a method called Weight Average Test-Time Adaptation (WATT) to improve test-time adaptation (TTA) for the CLIP model. The core idea is to use different text templates to construct multiple text prompts and adapt the model weights using these different prompts. During the evaluation stage, the prompts provide predictions based on the adapted weights and text embedding averaging across multiple prompts. The authors conduct comprehensive experiments that demonstrate the generalization and strong performance of the WATT method.

**Strengths:**

* The idea is simple but works effectively.
* Comprehensive experiments support the effectiveness of this method in the following aspects:
Robustness, as reflected in Table 6.
Efficient adaptation, as reflected in Table 4.
Generalization and state-of-the-art comparison, as shown in Table 7.

**Weaknesses:**

There are no clear weaknesses, but I do have some questions and potential limitations that do not affect my rating.

**Questions:**

* Is it possible to combine text prompt augmentation (this method) with TTA strategies that use image augmentation to achieve better results?
Some templates may be incorrect. For example, as shown in Table 2, prompt T0: "a photo of a {class k}" and prompt T2: "a bad photo of the {class k}" refer to the same image, but why is this photo considered bad? If you add distortion or corruption, then it might be appropriate to call it bad. Also, for T6: "art of the {class k}," if you use diffusion to generate an art image of the original image, then the prompt would match.
* Can you measure the similarity score of these text templates in the CLIP text encoder space?

**Limitations:**

Can the authors test whether this method can also improve the latest VLM, such as SigLIP?

---

> ### Author Rebuttal · Authors · 2024-08-07
>
> ***Q:* Is it possible to combine text prompt augmentation (this method) with TTA strategies that use image augmentation to achieve better results? Some templates may be incorrect. For example, as shown in Table 2, prompt T0: "a photo of a {class k}" and prompt T2: "a bad photo of the {class k}" refer to the same image, but why is this photo considered bad? If you add distortion or corruption, then it might be appropriate to call it bad. Also, for T6: "art of the {class k}," if you use diffusion to generate an art image of the original image, then the prompt would match.**
>
> *A:* To clarify how we chose the templates for weight averaging: The CLIP paper identifies 80 templates that enhance model robustness and performance. They ultimately conclude that 7 of these templates best summarize their model (see https://github.com/openai/CLIP/blob/main/notebooks/Interacting_with_CLIP.ipynb). In our work, we use these 7 generic templates and add the common one, “a photo of {},” based on their optimization. These templates are not specifically linked to the content of the images. However, linking these text templates to the images through augmentation or generation is a promising idea for future work. Thank you for the suggestion.
>
> ***Q:* Can you measure the similarity score of these text templates in the CLIP text encoder space?**
>
> *A:* As suggested, we computed the similarity between the text templates for each class and averaged them into a single matrix, which we have included in the PDF.
> The similarity between different templates for the same class can vary. This highlights that utilizing diverse templates, despite their individual variations, provides a richer set of information that significantly enhances the model's performance.
>
> ***Q:* Can the authors test whether this method can also improve the latest VLM, such as SigLIP?**
>
> *A:* Thank you for your suggestion! We have incorporated SigLip into our code and conducted a comparative analysis with our Sequential method (WATT-S) on different datasets. The results are summarized in the table below:
>  |  Dataset                | **SigLip** | **WATT-S** |
>  | :-------------------- | :-------------------- | :-------------------- |
>  | **CIFAR-10** | 66.35 | 75.02 &plusmn; 0.05|
>  | **CIFAR-10.1** | 57.30 | 65.87 &plusmn; 0.21|
>  | **CIFAR-10-C** | 37.52 | 45.29 &plusmn; 0.13|
>  | **CIFAR-100** | 33.97 | 65.87 &plusmn; 0.21|
>  | **CIFAR-100-C** | 14.43 | 20.05 &plusmn; 0.05|
>
> Our method, when applied to SigLip, shows significant improvements across all datasets, highlighting the effectiveness of WATT in enhancing performance.

---

> > ### Comment · Reviewer_wUmo · 2024-08-11
> > **Reply to authors**
> >
> > Thank you for your rebuttal. All my concerns have been addressed. The results reflected in SigLip are very positive, so I have increased my score.

---

> > > ### Author Response · Authors · 2024-08-12
> > >
> > > We're grateful for your detailed feedback and pleased that our response effectively addressed your concerns. The improvement with SigLip is encouraging, and we sincerely thank you for suggesting that comparison. Your input has been invaluable in strengthening our work.

---

### Author Rebuttal · Authors · 2024-08-07

We greatly appreciate the reviewers' insightful and constructive comments and are pleased to note that all three reviewers voted towards acceptance. We are also encouraged by the feedback highlighting the robustness and generalizability of our method (Reviewer wUmo), its superior performance across various domains (Reviewer y4Me), and the rigor of our experiments (Reviewer HDdi).

Below, we address all the questions raised by the reviewers, including additional baseline evaluations and clarifications as requested. For Reviewer wUmo’s convenience, we have included additional plots and results in the updated PDF.

---

###  **Continuation of Answer to Reviewer HDdi:**

***Q:* Have you considered evaluating WATT on real-world datasets with more complex domain shifts?**

*A:* Thank you for this remark suggesting that a better explanation of the datasets should be added to our supplementary materials. In our investigation, we use the VisDA-C dataset, which challenges models with two distinct domain shifts: the simulated shift and the video shift. The simulated shift includes 152,397 3D-rendered images across 12 diverse classes, while the video shift comprises 72,372 YouTube video frames spanning the same categories. This dataset addresses the diversity of imagery types applicable to a model, posing a significant challenge.

Moreover, we evaluate our proposed method on three other datasets: PACS, VLCS, and OfficeHome. These datasets help understand various domain shifts, including texture and style variations. The PACS dataset consists of 9,991 images across four domains (Art, Cartoons, Photos, Sketches) and seven classes. The VLCS dataset contains 10,729 images across four domains (Caltech101, LabelMe, SUN09, VOC2007) and five classes. Lastly, the OfficeHome dataset includes 15,588 images across four domains (Art, Clipart, Product, Real) and 65 classes. Evaluating across these distinct domain shifts showcases the generalizability of our method.
These datasets are more representative of real-world scenarios compared to CIFAR, with complex domain shifts.

---

### Decision · Program_Chairs · 2024-09-25

**Decision:**

Accept (poster)

**Comment:**

The paper proposes "Weight Average Test Time Adaptation" (WATT) for VLMs like CLIP, aimed at enhancing their ability to handle domain shifts during inference without retraining. The method leverages multiple text prompt templates to adapt model weights at test time and aggregates predictions using weight averaging to improve performance across datasets with domain shifts, such as CIFAR and VisDA. The results show notable improvements in robustness and generalization without the need for model modifications or additional training.

The paper is technically solid and presents a practical approach to improving VLMs under domain shifts without retraining. Despite the limited novelty, the experimental results demonstrate benefits in terms of robustness and generalization, making it a valuable contribution to the test-time adaptation field. While there are some concerns about the breadth of evaluation and computational aspects, the overall impact on the VLMs and TTA subfield justifies its acceptance.